# Value-complexity tradeoff explains mouse navigational learning

**Nadav Amir**[1], **Reut Suliman-Lavie**[2], **Maayan Tal**[2], **Sagiv Shifman**[2], **Naftali Tishby**[1,3☉], **Israel Nelken**[1,2☉]*

**1** Edmond and Lily Safra Center for Brain Sciences (ELSC), Hebrew University, Jerusalem, Israel, **2** The Alexander Silberman Institute of Life Sciences, Hebrew University, Jerusalem, Israel, **3** The Benin School of Computer Science and Engineering, Hebrew University, Jerusalem, Israel

☉ These authors contributed equally to this work.
* israel.nelken@mail.huji.ac.il

## Abstract

We introduce a novel methodology for describing animal behavior as a tradeoff between value and complexity, using the Morris Water Maze navigation task as a concrete example. We develop a dynamical system model of the Water Maze navigation task, solve its optimal control under varying complexity constraints, and analyze the learning process in terms of the value and complexity of swimming trajectories. The value of a trajectory is related to its energetic cost and is correlated with swimming time. Complexity is a novel learning metric which measures how unlikely is a trajectory to be generated by a naive animal. Our model is analytically tractable, provides good fit to observed behavior and reveals that the learning process is characterized by early value optimization followed by complexity reduction. Furthermore, complexity sensitively characterizes behavioral differences between mouse strains.

## Author summary

Goal directed learning typically involves the computation of complex sequences of actions. However, computational frameworks such as reinforcement learning focus on optimizing the reward, or value, associated with action sequences while ignoring their complexity cost. Here we develop a complexity-limited optimal control model of the Morris Water Maze navigation task: a widely used tool for characterizing the effects of genetic and other experimental manipulations in animal models. Our proposed complexity metric provides new insights on the dynamics of navigational learning and reveals behavioral differences between mouse strains.

## Introduction

Adaptive decision-making is often modeled, within the framework of reinforcement learning, as a process of generating actions associated with a high expected reward signal, or value (or low negative value, also called cost) [1]. Learning is described within this framework as a

**Data Availability Statement:** The numerical data of all swimming trajectories, as well as Matlab files for generating the figures are available on the Open Science Framework repository, at https://osf.io/3wgzx/ or DOI 10.17605/OSF.IO/3WGZX.

**Funding:** This study was supported by Advanced European Research Council (ERC) grant 340063 (project RATLAND; https://erc.europa.eu/) and F.I. R.S.T. grant 1075/13 to IN, and by a personal grant from the Israel Science Foundation (https://www.isf.org.il/) grant no. 575/17 to SS. The funders had no role in study design, data collection and analysis, decision to publish, or preparation of the manuscript.

**Competing interests:** The authors have declared that no competing interests exist.

process of finding a sequence of actions which maximizes the cumulative value (or minimizes the cumulative cost). The rule by which actions are selected is sometimes called a policy. The value (or cost) typically represents some notion of task performance (for example the total time for task completion). Importantly, while the value (or cost) can be often interpreted in terms of task performance level and energy expenditure associated with a policy, it ignores internal information processes involved in generating or computing the policy, which may have their own cost. We show here that information processing costs have important consequences to learning. Thus, biological models of behavior need to quantify policies not only in terms of their expected rewards but also in terms of their information processing costs.

To address this fundamental issue, we develop a framework for describing biological learning as a trade-off between two measures: *value*, which reflects task performance level as well as energetic or metabolic constraints, and *complexity*, which relates to the internal information processing limitations of the organism. Learning is thus formalized as a constrained optimization problem: maximizing value under a given complexity constraint, or equivalently, minimizing complexity under a given value constraint.

We demonstrate the usefulness of this framework by developing a complexity-limited, control-theoretic model of a mouse navigating a large, circular tank of water; the so-called Morris Water Maze navigation task. The Morris Water Maze is widely used in neuroscience for studying cognitive processes and neural mechanisms underlying spatial learning and memory [2]. Because of its simplicity and robustness, it is used to characterize the effects of many different experimental manipulations, including genetic modifications, manipulation of brain activity through lesions or opto- and chemogenetics, behavioral manipulations, and drugs. Efficient metrics of behavior in the Morris Water Maze are therefore of great importance. The task involves placing animals, typically rats or mice, at one of a number of possible starting locations in a large circular tank filled with opaque water, from which they can escape by reaching a submerged platform whose location is fixed (Fig 1). The goal of the animal is to learn the location of the platform. The animal can use distal visual cues such as high contrast images placed on the walls of the room, which are fixed and consistent from trial to trial. Initially, the animal tends to swim near the walls of the tank, a behavior known as thigmotaxis, but shortly after learning the location of the platform, the animal starts taking shorter and more direct swimming paths towards it. Task performance is typically quantified using latency to platform, path length, the proportion of time spent in the quadrant of the tank in which the platform is positioned, or the average distance to the platform while swimming [3].

We construct a model of the water maze in three steps. First, we describe the physical properties of a naive mouse, i.e., a mouse who is not aware of the existence and location of the platform. Such mice tend to swim in meandering, quasi-circular trajectories near the tank walls. We therefore model their motion using a stochastic, damped harmonic oscillator. Second, we compute the optimal trajectories from each starting point in the tank to the stationary platform using a classical result from linear optimal control theory: the Linear Quadratic Regulator, or LQR [4]. These theoretically derived trajectories are optimal in the sense that they optimize a value functional over the feasible trajectories. In the case of the LQR, the value functional is a quadratic form that is related to the total distance travelled as well as to the forces needed in order to reach the platform.

The main theoretical contribution of this paper consists of the third step, in which we account for the gradual learning process by augmenting the LQR value functional with a complexity functional that measures the difference, in a statistical sense, between the actions generating the winding trajectories of naive mice and those generating the more direct trajectories of the trained ones. Our complexity measure is taken as a fundamental quantifier of the computational cost involved in action generation and selection. It is not meant to serve as a

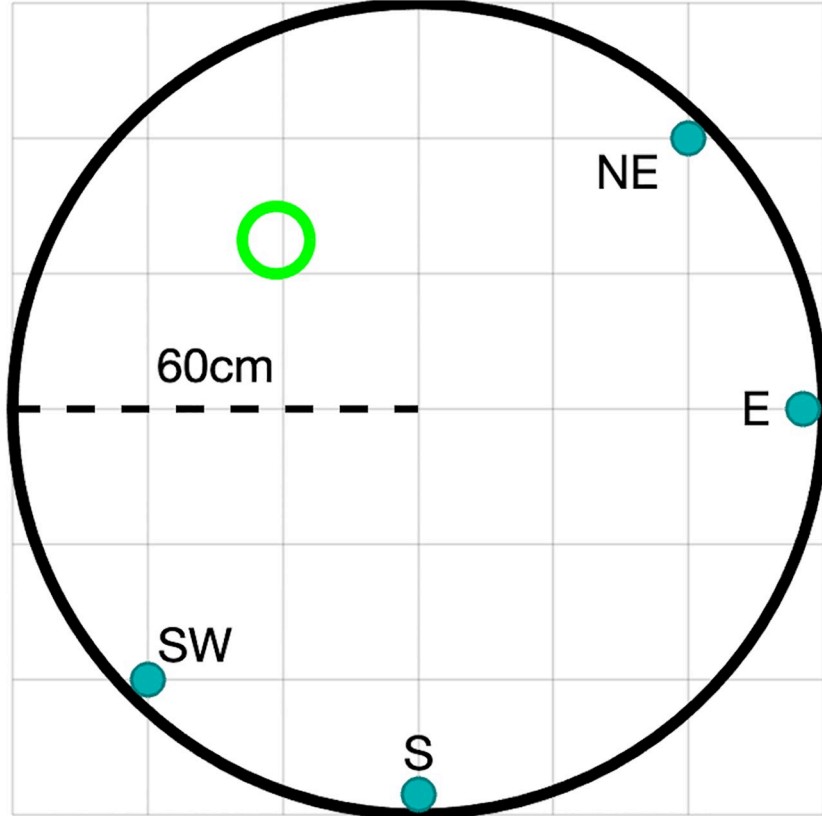

**Fig 1. The water maze experiment.** Schematic figure of the water maze experiment. The fixed platform is shown in green. Release locations are indicated near the tank's perimeter.

proxy for muscle activation or some such metabolic cost, which are supposed to be indexed by the value functional.

The choice of our measure of complexity is based on the following considerations. The behavior of naive mice, who know nothing about the location of the platform, should have the lowest complexity by definition. The behavior of trained mice, on the other hand, whose trajectories are often shorter and more direct, incurs high complexity cost because in the context of the model, which includes damping and noise, it requires the moment-by-moment generation of precise motor commands to counteract damping and correct precisely for the noise. This would presumably result in a higher computational and cognitive load. Thus, the complexity is not that of the swimming trajectories themselves, but rather that of the goal-directed computational processes needed to generate them. Our complexity measure captures the fact that the motor commands executed by trained mice are unlikely to be generated by naive mice. The specific form of our complexity measure is based on a result from large deviations theory known as Sanov's theorem [5] (see Large deviations theory and Sanov's theorem in the Methods section for details). In the context of the model, Sanov's theorem implies that our complexity measure quantifies how (un)likely it is for a sequence of actions (motor commands), generated by a mouse that already learned something about the water maze, to be generated by a naive mouse.

Using these two measures, value and complexity, we carry out an analysis of the trajectory learning process in the Morris water maze task. This analysis provides two interesting results: first, it shows that wildtype mice, in particular females, initially tend to optimize the value of

the paths by finding shorter paths to the platforms, and only later start reducing the complexity of the paths by finding simpler trajectories without reducing value. Second, complexity is sensitive to subtle features of the trajectories which are undetected by standard water maze performance measures, and can be used to characterize important behavioral differences between mouse strains.

## Results

### Modeling the water maze

**The naive mouse.** In the water maze task, mice learn the location of a submerged platform within a water tank, using mostly visual cues. Briefly, mice were placed facing the tank wall at one of four start locations designated as East (E), South (S), Northeast (NE) and Southwest (SW) directions, whereas the platform remained fixed in the middle of the Northwest (NW) quadrant (Fig 1). Over a period of four consecutive days, each mouse was released four times every day, once from each starting location in a randomized order. If a mouse did not find the platform within 60 seconds, it was positioned by the experimenter on the platform and left there for an additional 30 seconds, allowing it to orient itself in relation to distal visual cues on the walls of the tank and the room.

We first modelled the dynamics of a naive mouse as it swims around the circular tank. For our purposes, a naive mouse is one which has no experience in the water maze task and in particular does not have any information about the location of the platform. Since naive mice tend to move near the tank perimeter, with long segments that are roughly circular, we used a 2-D stochastic harmonic oscillator to model their motion. We added a damping term to model water viscosity and additive Gaussian noise to allow for randomness in the trajectories. We refer to this model of mouse motion as the naive or *uncontrolled* mouse model as it does not contain any information about the location of the platform. It can be expressed in state-space notation by the following linear-time-invariant stochastic dynamical system:

$$\dot{\mathbf{x}}(t) = A\mathbf{x}(t) + \xi(t), \tag{1}$$

where $\mathbf{x}(t)$ is the 4-dimensional state of the mouse (bold characters represent vectors throughout),

$$\mathbf{x}(t) = [\mathbf{q}(t), \mathbf{p}(t)]^T, \tag{2}$$

and $\mathbf{q}(t) = (q_x(t), q_y(t))$, $\mathbf{p}(t) = (p_x(t), p_y(t))$ are the position and velocity coordinates of the mouse respectively, i.e., $\mathbf{p}(t) = \dot{\mathbf{q}}(t)$. The definition of state in terms of both position and velocity coordinates enables us to describe the Newtonian dynamics of the damped harmonic oscillator by the following matrix:

$$A = \begin{pmatrix} 0 & 0 & 1 & 0 \\ 0 & 0 & 0 & 1 \\ -k/m & 0 & -\gamma/m & 0 \\ 0 & -k/m & 0 & -\gamma/m \end{pmatrix} \tag{3}$$

with the two parameters $k$ and $\gamma$ representing the restoring force and damping constants respectively. For simplicity, we use a constant mass of $m = 20g$ (typical to mice) and further assume, due to circular symmetry, that $k,\gamma$ are isotropic (equal $x$ and $y$ components). The noise term, $\xi(t) \sim \mathcal{N}(\mathbf{0}, \Sigma_\xi)$, is a zero mean, stationary Gaussian process with covariance matrix $\Sigma_\xi$.

The noise components of the position and velocity are also assumed to be isotropic, due to circular symmetry, and independent of each other, so that the noise covariance matrix has the following diagonal form:

$$\Sigma_\xi = \begin{pmatrix} \sigma_q^2 & 0 & 0 & 0 \\ 0 & \sigma_q^2 & 0 & 0 \\ 0 & 0 & \sigma_p^2 & 0 \\ 0 & 0 & 0 & \sigma_p^2 \end{pmatrix},$$

(4)

Where $\sigma_q^2$ and $\sigma_p^2$ represent position and velocity noise variances respectively.

In summary, we construct a linear, time-invariant, stochastic dynamical system model for the motion of naive mice in the water maze that has four parameters: $k$, $\gamma$, $\sigma_q$ and $\sigma_p$.

**Modeling optimal behavior.** To model the learned behavior of the mouse at the end of training, we add a *control signal* term, $\mathbf{u}(t) = [u_x(t), u_y(t)]^T$, to the free model described above (Eq 1). This term describes the forces exerted by the mouse to navigate toward the platform. The resulting control system, which we refer to as the *controlled* model, can be described as follows:

$$\dot{\mathbf{x}}(t) = A\mathbf{x}(t) + B\mathbf{u}(t) + \xi(t),$$

(5)

where the matrix:

$$B = \begin{pmatrix} 0 & 0 & 1/m & 0 \\ 0 & 0 & 0 & 1/m \end{pmatrix}^T,$$

(6)

aligns the control components, $u_x(t)$ and $u_y(t)$, with the corresponding mouse acceleration coordinates, $\dot{p}_x(t)$ and $\dot{p}_y(t)$, in line with their role as forces exerted by the mouse.

The problem of finding the optimal behavior is now reduced to the selection of a good control signal $\mathbf{u}(t)$. To define what we mean by that, we introduce a cost—a measure that takes into account those features of the task that require energy expenditure from the mouse. The cost is a functional of the control signal (applied forces) as well as of the resulting swimming path. We then define the value functional as the negative cost. This somewhat circuitous definition is required since control theory typically uses cost, while studies of animal behavior usually use value. Since cost and value are equivalent up to sign, we will use the two terms interchangeably from here on, while mostly preferring the use of value.

Formally, we define a functional, $J(\mathbf{x}(t), \mathbf{u}(t))$, as the integrated "energetic cost" of the trajectory. Here $\mathbf{x}(t)$ is the swimming path that results from the application of the force $\mathbf{u}(t)$. Once $J(\mathbf{x}(t), \mathbf{u}(t))$ is specified, optimal control theory provides the force to apply at each moment in time in order to steer the animal to the platform at minimal cost. Any other control signal will result, on average, in costlier trajectories.

We use a quadratic cost functional with three terms representing different factors that contribute to deviations from optimal behavior:

$$J[\mathbf{x}(t), \mathbf{u}(t)] = \mathbb{E}\Big[\frac{1}{2}\int_0^T \Big((\mathbf{x}(t) - \bar{\mathbf{x}})^T Q(\mathbf{x}(t) - \bar{\mathbf{x}}) +$$

$$\mathbf{u}(t)^T R\mathbf{u}(t) + 2(\mathbf{x}(t) - \bar{\mathbf{x}})^T N\mathbf{u}(t)\Big) dt\Big],$$

(7)

where $T$ denotes the duration of the trial. The matrices $R$ and $Q$ are assumed to be positive

definite and semi-definite respectively, and $\bar{\mathbf{x}} = [\bar{q}_x, \bar{q}_y, 0, 0]^T$ are the (fixed) state space coordinates of a stationary mouse on the platform.

The first term in Eq 7 represents the cost of distance, in state-space, from the target, effectively giving a higher value to trajectories which reach the platform faster and remain closer to it. The second term represents a penalty on force exertion, since reaching the target using less force is energetically desirable. The mixed third term can account for possible interactions between position and force exertion; e.g., the same force may be less desirable when the animal is near the target compared to when it is near the perimeter.

The integral in Eq 7 is calculated along the trajectory, from the release location of the animal into the arena and until it either reaches the platform or is positioned there by the experimenter. The expectation is taken over all possible realizations of the additive Gaussian noise process $\xi(t)$ in Eq 5.

We can now define the value functional simply as the negative cost:

$$V[\mathbf{x}(t), \mathbf{u}(t)] = -J[\mathbf{x}(t), \mathbf{u}(t)]. \tag{8}$$

While the cost is always positive and is small when performance is good, the value is always negative and becomes large (close to 0) when performance is good.

Finding the optimal control signal which maximizes the value functional (Eq 8) under the model dynamics (Eq 5) is a classical problem in optimal control theory. Its solution is called the *Linear Quadratic Regulator* [4], a linear force proportional to the difference between the current and target states. In our case these correspond to the states of the animal and the platform respectively:

$$\mathbf{u}^{opt}(t) = -K(\mathbf{x}(t) - \bar{\mathbf{x}}), \tag{9}$$

where the superscript indicates that this is the *optimal* control signal; i.e., the one maximizing the value functional, Eq 8. The feedback gain matrix, $K$, is computed from the parameters of the problem—the matrices $A$ and $B$ that define the dynamics and the matrices $Q$, $R$ and $N$ that define the cost functional. The computation of $K$ is described in in Computing the optimal feedback gain in the Methods section with additional details provided in the mathematical appendices Derivation of the Riccati equation and Boundary conditions and transients.

To apply the model to the empirical trajectory data, which was sampled at a rate of $\Delta t = 0.2s$, we transform it into a discrete-time form. The discrete-time dynamics that approximates Eq 5 can be written as:

$$\mathbf{x}_{t+1} = A_{\Delta t}\mathbf{x}_t + B_{\Delta t}\mathbf{u}_t + \xi_t , \tag{10}$$

where $\xi_t \sim \mathcal{N}(0, \Sigma_{\Delta t})$ and $A_{\Delta t}$, $B_{\Delta t}$ and $\Sigma_{\Delta t}$ denote the discrete-time equivalents of $A$, $B$ and $\Sigma_\xi$ respectively. They can be computed from their continuous-time counterparts, detailed in Model discretization in the Methods section. The discrete-time version of the cost functional (Eq 7) can be written as:

$$J_{\Delta t}[\mathbf{x}_t, \mathbf{u}_t] = \mathbb{E}\Big[\frac{1}{2}\sum_{t=1}^{T}\Big((\mathbf{x}_t - \bar{\mathbf{x}})^T Q_{\Delta t}(\mathbf{x}_t - \bar{\mathbf{x}})+$$
$$\mathbf{u}_t^T R_{\Delta t}\mathbf{u}_t + 2(\mathbf{x}_t - \bar{\mathbf{x}})^T N_{\Delta t}\mathbf{u}_t\Big)\Big], \tag{11}$$

where $T$ denotes here the number of samples along the path and $Q_{\Delta t}$, $R_{\Delta t}$ and $N_{\Delta t}$ can be computed from their continuous-time counterparts, as detailed in Model discretization in the

Methods section. The corresponding value is again defined simply as the negative cost:

$$V_{\Delta t}[\mathbf{x}_t, \mathbf{u}_t] = -J_{\Delta t}[\mathbf{x}_t, \mathbf{u}_t]. \tag{12}$$

The solution to the discrete-time optimal control problem, maximizing the discrete-time value functional (Eq 12), is given by:

$$\mathbf{u}_t^{opt} = -K_{\Delta t}(\mathbf{x}_t - \bar{\mathbf{x}}), \tag{13}$$

where the discrete-time feedback gain matrix, $K_{\Delta t}$, can be computed from the discrete-time dynamics and cost matrices, as detailed in Computing the optimal feedback gain in the Methods section.

**Modeling the learning: Complexity constrained control.**   So far we described a standard optimal control problem, consisting of finding the control signal which generates trajectories that maximize the value functional, Eq 12. Such models are widely used in fields such as aircraft and naval navigation and control. Biological organisms, however, are subject not only to performance and energetic (metabolic) limitations, but also to complexity, or information processing constraints. These constraints include memory as well as the information processing limitations involved in sensing and acting. Therefore, we introduce a measure of complexity that quantifies the information required for action selection. This measure is defined by comparing two actions, the one selected by the current policy and a default action that corresponds to the choices of a naive animal (that does not know where the platform is located). Under this definition, the complexity of a sequence of actions increases as the trajectory it generates becomes increasingly distinguishable from a naive one.

Formally, we introduce an additional functional, $I_{\Delta_t}[\mathbf{x}_t, \mathbf{u}_t]$, representing the complexity of a trajectory generated by a given control signal $\mathbf{u}_t$:

$$I_{\Delta_t}[\mathbf{x}_t, \mathbf{u}_t] = \frac{1}{2}\sum_{t=1}^{T-1} D_{KL}\big(P(\mathbf{x}_{t+1} \mid \mathbf{x}_t, \mathbf{u}_t)\|P(\mathbf{x}_{t+1} \mid \mathbf{x}_t)\big), \tag{14}$$

where $D_{KL}(P(\mathbf{x}_{t+1}|\mathbf{x}_t, \mathbf{u}_t) \| P(\mathbf{x}_{t+1}|\mathbf{x}_t))$ is the Kullback-Leibler (KL) divergence between the state transition likelihood of the trajectory $\mathbf{x}_t$, under the control signal $\mathbf{u}_t$:

$$P(\mathbf{x}_{t+1} \mid \mathbf{x}_t, \mathbf{u}_t) \sim \mathcal{N}(A_{\Delta t}\mathbf{x}_t + B_{\Delta t}\mathbf{u}_t, \Sigma_{\Delta t}), \tag{15}$$

and the state transition likelihood of the same trajectory under the free (uncontrolled) model:

$$P(\mathbf{x}_{t+1} \mid \mathbf{x}_t) \sim \mathcal{N}(A_{\Delta t}\mathbf{x}_t, \Sigma_{\Delta t}). \tag{16}$$

The KL divergence, also called relative entropy, is a measure of the difference, in information theoretic terms, between two probability distributions [6]. The KL divergence between two discrete distributions, $P_1$ and $P_2$, is defined as follows:

$$D_{KL}(P_1\|P_2) = \sum_x P_1(x) \log \frac{P_1(x)}{P_2(x)}. \tag{17}$$

It is non-negative and equals zero only when the two distributions are almost everywhere equal. Our use of the KL divergence as a measure of complexity is based on a result from large deviations theory known as Sanov's theorem [5]. In our context, Sanov's theorem states that the likelihood of a naive trajectory to achieve a certain value is determined by the minimal obtainable KL divergence between a controlled trajectory distribution that achieves that value and the naive trajectory distribution. Furthermore, controlled trajectories with a larger KL divergence are exponentially less likely to occur under the naive behavior. Thus, our

complexity functional, Eq 14, is a natural measure of how (un)likely is it for a particular controlled behavior to occur with respect to the naive distribution. While it is possible to provide a continuous time version of complexity, at least in the Gaussian noise case, it is simpler and more transparent in the discrete case which is anyway what we computed on the empirical data (see Computing the theoretical value-complexity curves in the Methods section for details).

We combine the complexity with the value functional (Eq 12), weighted by a non-negative parameter $\beta$, to obtain the following so-called *free energy* functional:

$$F_{\Delta t}[\mathbf{x}_t, \mathbf{u}_t, \beta] = I_{\Delta t}[\mathbf{x}_t, \mathbf{u}_t] - \beta V_{\Delta t}[\mathbf{x}_t, \mathbf{u}_t]. \tag{18}$$

The negative sign of the value is introduced since we will eventually minimize, rather than maximize, the free energy. By analogy to statistical physics, the non-negative Lagrange multiplier $\beta$ plays a role analogous to inverse temperature in thermodynamic free energy [7].

The complexity constrained optimal control problem consists of finding, for any value of $\beta$, the control signal which minimizes the free energy functional, Eq 18, under the model dynamics, Eq 10. Minimizing the free energy functional prescribes the optimal trade-off, determined by $\beta$, between low complexity, i.e., minimizing the complexity term $I_{\Delta t}[\mathbf{x}_t, \mathbf{u}_t]$, and high value, i.e., maximizing the value term $V_{\Delta t}[\mathbf{x}_t, \mathbf{u}_t]$. Thus, minimizing the free energy is equivalent to finding the simplest paths that achieve the value given by $V_{\Delta t}[\mathbf{x}_t, \mathbf{u}_t]$. These paths are simplest in the sense of minimizing $I_{\Delta t}[\mathbf{x}_t, \mathbf{u}_t]$, that is, they are the most similar to the free swimming paths. Alternatively, the solution is equivalent to maximizing the value $V_{\Delta t}[\mathbf{x}_t, \mathbf{u}_t]$ among all paths whose complexity is constrained to a given level $I_{\Delta t}[\mathbf{x}_t, \mathbf{u}_t]$.

When $\beta \approx 0$, corresponding to high thermodynamic temperatures, the free energy consists of the complexity term only, and the optimal solution is close to the naive swimming behavior (which minimizes the complexity by definition). Conversely, when $\beta$ is very large, corresponding to low temperatures in the thermodynamic analogy, the complexity term becomes negligible and the optimal solution becomes the optimal control solution of the original LQR problem, maximizing the value. For intermediate $\beta$ values, the trajectories that minimize the free energy represent a balance between minimization of complexity and maximization of value.

Importantly, the complexity constrained optimal control for a given $\beta$ value, obtained by minimizing the free energy (Eq 18) subject to the dynamics (Eq 10), results in a linear feedback control signal:

$$\mathbf{u}_t^\beta = -K_{\Delta t}^\beta (\mathbf{x}_t - \bar{\mathbf{x}}), \tag{19}$$

where the optimal feedback gain matrix, $K_{\Delta t}^\beta$, now depends on $\beta$ (see Computing the optimal feedback gain in the Methods section).

**Fitting the model to data.** We applied the model to swimming paths from wildtype mice and mice with a heterozygous mutation in the *Pogz* gene (pogo transposable element-derived protein with zinc finger domain). Heterozygous loss-of-function mutations in the human *POGZ* gene are associated with intellectual disability and autism spectrum disorder. See Experimental procedures in the Methods section for more details about these mice.

The parameters of the model were estimated from the behavior of the wildtype mice data in three steps, described in detail in subsection Estimating model parameters of the Methods section. First, we estimated the parameters of the free model (Eq 1) using the first swimming trial of each mouse. Next, we estimated the parameters of the value functional (Eq 8). This time the data consisted of the last swimming trial of each mouse. Finally, we estimated the value of the learning parameter $\beta$, using the rest of the swimming paths, grouped by day.

The free model parameters, estimated using maximum likelihood from the first trials of each wildtype mouse, were as follows:

$$(\hat{k}, \hat{\gamma}, \hat{\sigma}_q, \hat{\sigma}_p) = (3.7g/cm^2, 0.47g/cm, 1.1cm/s, 4.5cm/s^2). \tag{20}$$

The estimated harmonic oscillator is stable and underdamped, with a damping coefficient of:

$$\zeta = \gamma/(2\sqrt{mk}) \approx 0.03, \tag{21}$$

and an angular frequency of:

$$\omega = \omega_0\sqrt{1 - \zeta^2} \approx \sqrt{k/m} \approx 0.43rad/s, \tag{22}$$

where $\omega_0 = \sqrt{k/m}$ is the undamped angular frequency.

Next to the wall ($r = 60cm$), these estimates imply a swimming speed of $v = \omega r \approx 26cm/s$, in agreement with typically reported mean swimming speeds for mice [8, 9].

The value functional weight matrices, $Q$, $R$ and $N$, were estimated using the final trials of each wildtype mouse. Since many weight matrices can result in the same steady state feedback gain, and therefore in the same trajectory, we estimated the feedback gain matrix $K$ directly, and used it to infer a particular choice of $Q$,$R$ and $N$ matrices corresponding to that feedback gain (see Estimating model parameters in the Methods section for details). We initially estimated the 2 × 4 feedback gain matrix $K$ with no constraints on its entries. This yielded the following non-parametric maximum likelihood estimate $\hat{K}_{NP}$:

$$\hat{K}_{NP} = \begin{pmatrix} 0.21 & -0.07 & 0.22 & -0.30 \\ -0.02 & 0.17 & 0.32 & 0.33 \end{pmatrix}. \tag{23}$$

The structure of this matrix led us to specify the following parametric form for $K$, reducing the number of free parameters from 8 to 3:

$$K = \begin{pmatrix} K_r & 0 & K_t\cos(K_\alpha) & -K_t\sin(K_\alpha) \\ 0 & K_r & K_t\sin(K_\alpha) & K_t\cos(K_\alpha) \end{pmatrix}, \tag{24}$$

The maximum likelihood value for the parametric form of $K$ was:

$$\hat{K} = \begin{pmatrix} 0.20 & 0 & 0.24 & -0.29 \\ 0 & 0.20 & 0.29 & 0.24 \end{pmatrix}, \tag{25}$$

corresponding to the fitted parameters $\hat{K}_r = 0.20, \hat{K}_t = 0.38, \hat{K}_\alpha = 0.89$. The structure of $\hat{K}$ indicates that the force applied by the mouse can be decomposed into a radial component that drives it towards the platform and a tangential one that counteracts the tendency to rotate around the center of the tank. The ratio $K_r/K_t = 0.53$ describes the relative magnitude of the radial force component with respect to the tangential one. The log-likelihood values of the two versions of $K$ were similar; namely −1.5983 for $\hat{K}$ and −1.5992 for $\hat{K}_{NP}$.

We estimated the maximum likelihood $\beta$ values for each mouse over the 4 training days using all paths of that mouse on each day, excluding the ones used for estimating the model parameters; i.e., the first path on the first day and the last path on the last day (see Estimating $\beta$ in the Methods section for details). The resulting mean $\beta$ values for the four days were $\beta_1 = 0.22, \beta_2 = 4.5, \beta_3 = 36.8$ and $\beta_4 = 475$, for days 1-4 respectively. Thus, the $\beta$ parameter increased by a factor of roughly 10 from day to day, even from day 3 to day 4, when the latency to platform largely saturated (see below). This observation suggests that the learning process in the

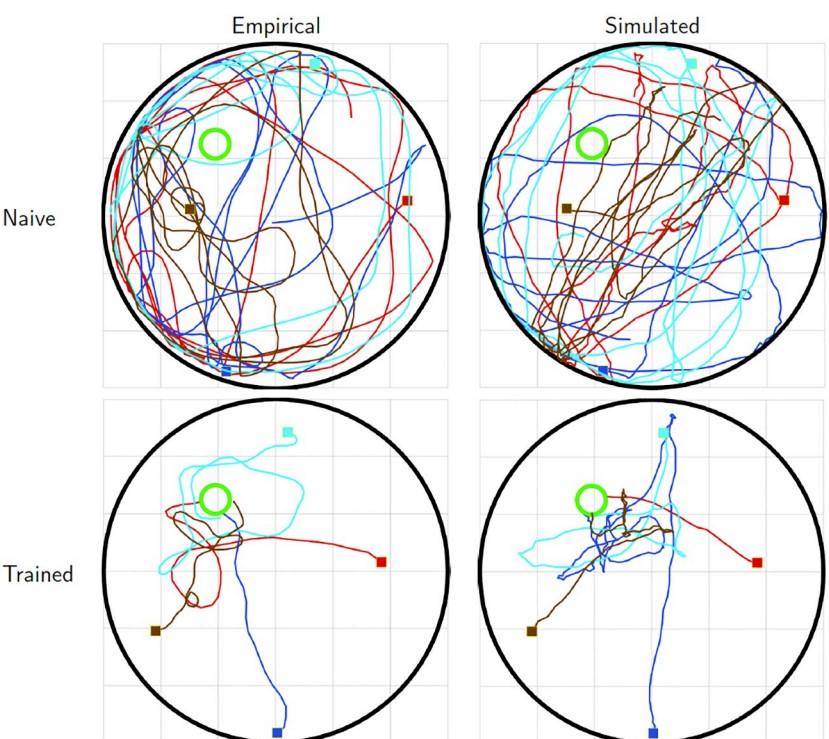

**Fig 2. Empirical and model generated trajectories.** Top: empirical trajectories generated by naive (day 1) mice (left) and simulated trajectories generated by the uncontrolled model (right). Bottom: empirical trajectories generated by trained (day 4) mice (left) and simulated trajectories generated by the optimal control model (right). Initial positions, indicated by filled squares, and velocities, were matched between empirical and simulated trajectories. Trajectories simulated by the uncontrolled model are confined to tank boundaries.

water maze is richer and more intricate than suggested by the standard performance features such as latency to platform.

**Model validation.** To illustrate the properties of the model fit, we simulated free ($\beta = 0$) and optimal ($\beta = \infty$) trajectories, and compared them to the empirical trajectories of naive and trained mice respectively. Fig 2 qualitatively compares empirical trajectories with typical trajectories generated by the model. Trajectories of naive mice are compared to trajectories generated by the uncontrolled model (top), and trajectories of trained mice with those generated by the LQR model to optimal control (bottom). The paths are not expected to be identical, since this would require the noise used in the simulation to match the unknown noise that presumably occurred during the actual experiment. Rather, the figure illustrates the comparable characteristics of the resulting swimming paths. For visualization purposes, the simulated trajectories of naive mice are subjected to a hard boundary condition at the perimeter of the tank. This boundary condition is not imposed by the model as this would introduce a hard non-linearity which would greatly complicate the analytical solution. Nevertheless, despite its simplicity, the model can reproduce both the quasi-circular meandering of the naive mice, and straighter, platform directed swimming paths that are typical of the trained mice.

We quantitatively compared the model-generated paths with the empirical values of several water maze performance measures. We considered trajectory duration (latency to platform), trajectory length, average velocity, and the mean distance to the platform during the trajectory (a learning measure sometimes referred to as the Gallagher Index [10]). We computed each of

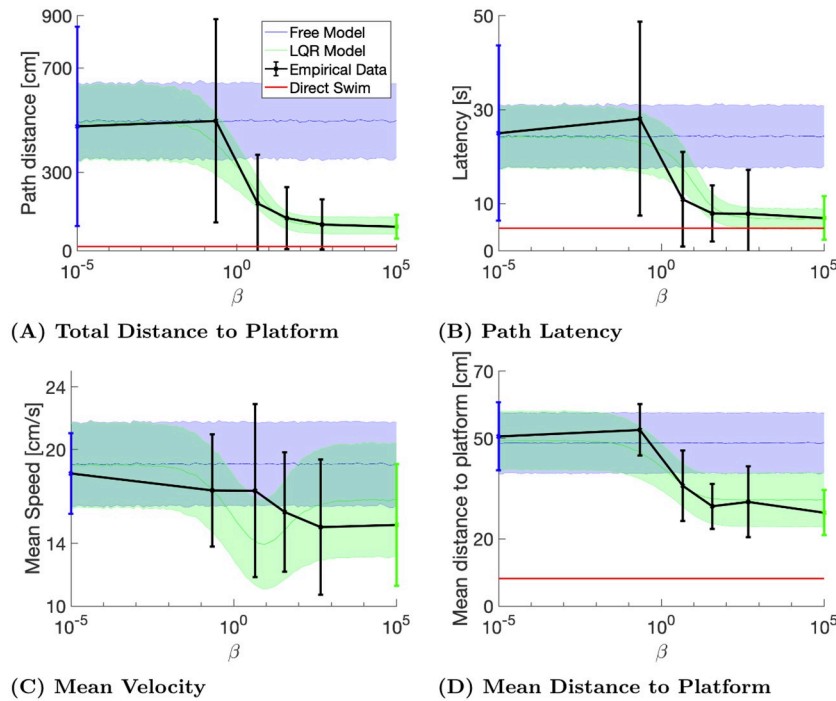

**Fig 3. Model and empirical performance measures.** The first and last empirical data points represent the trials used for training the uncontrolled (blue) and optimal control (green) models. The four mid points (black) represent the four training days. The empirical points shown are for the *E* release location. Error bars indicate standard deviations. The shaded areas represent one standard deviation above and below the average computed from the simulated swimming paths. The red line in panels A, B and D correspond to the minimum achievable value for the corresponding parameter, computed using a straight swimming path from the release location to the platform, using the mean velocity over all trials from the corresponding release location.

these these values at six time points (Fig 3): the first swimming path of each mouse (compared with the uncontrolled model, $\beta = 0$), the average for all the rest of the swimming paths on day 1, all swimming paths of day 2, all swimming paths of day 3, the swimming paths of day 4 except for the last one (compared with model swimming paths with the corresponding $\beta$ for each day), and the average value for all the last swimming paths of each mouse (compared with the controlled model, $\beta = \infty$).

For the experimental data, all of these values decreased throughout training. For the model (as a function of $\beta$), path length, latency and mean distance to the platform also decreased monotonically. Thus, $\beta$ behaves as a (single) learning parameter, representing the gradual transition from naive to trained navigation behavior. Quantitatively, the total distance to the platform (Fig 3a) was well estimated, presumably because the model was mostly fitted to distance data. The mean latency to the platform and the mean distance to the platform were also quite well estimated by the model. The mean velocity (Fig 3c) was less well estimated by the model, although the average measured values were still within one standard deviation of the average simulated values.

**Change of control during learning.** We next wanted to see how the learned control changed as function of $\beta$. Fig 4 shows a single empirical trajectory of a mouse released at the *S* starting location on the first day of training. For every 5th sample point along the trajectory, the actual velocity vector is shown in black and model predicted velocity vectors, for 75 linearly spaced $\beta$ values between $10^{-2}$ and $10^{2}$, are shown with a color scale representing the magnitude of $\beta$. As expected, higher $\beta$ values (shown as red and yellow arrows) resulted in predicted

 

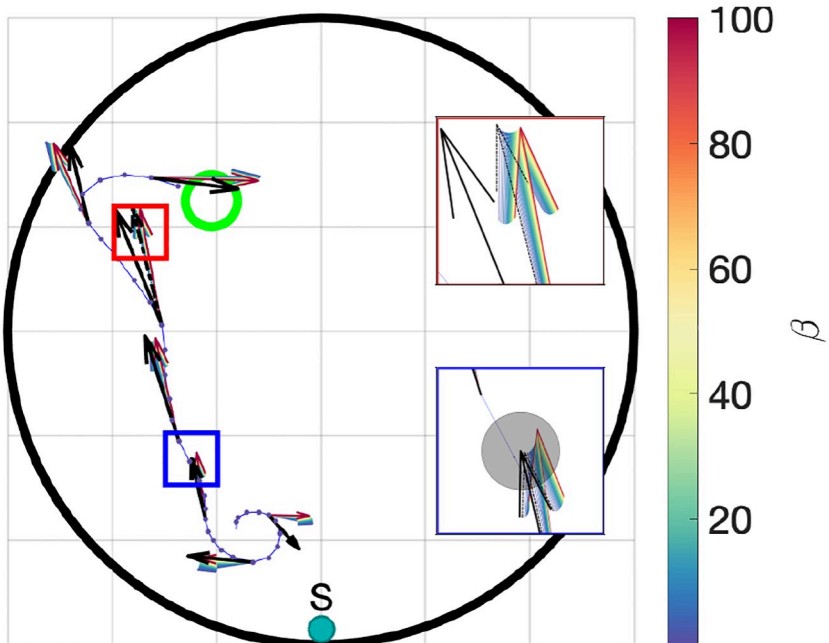

**Fig 4. Model predictions along a trajectory as a function of β.** Actual and model predicted vectors for different β values shown at several points along an empirical path from the first day starting at the *S* release location. The black arrows represent the actual velocity vectors at the same point. Model predicted vectors corresponding to large β values (red and yellow arrows), are better oriented towards the platform than the those corresponding to smaller β values (blue and green). The non-monotonic speed profile (arrow length) as a function of beta can be seen in the top inset (red border). The standard deviation of the velocity noise is shown as a grey circle around the tip of the predicted velocity vector in the bottom inset (blue border). The velocity vectors corresponding to the estimated value of β that best fits the data (β = 0.273) are indicated by dashed black arrows in the insets.

velocity vectors rotated towards the platform, compared to those predicted by lower β values (green and blue arrows). The velocity vector corresponding to the value of β that best fits the data are shown in the insets (dashed black arrows). The predicted velocity vectors represent expected values: the actual velocity vectors (solid black arrows) include the contribution of the noise, represented in the lower inset of Fig 4 by the gray circle. The actual velocity vectors were mostly consistent with those predicted by the model, although they tended to be closer to velocities corresponding to intermediate β values.

As suggested from Fig 3c, the lengths of the model velocity vectors were a non-monotonic function of β, decreasing for intermediate values and then increasing again for large β values (Fig 4 top inset). Thus, the model predicted that swimming speed would decrease first, then increase again as the mice converge upon the optimal control. This trend was not observed; rather, swimming speed decreased slightly on average between the first and last day.

**Value and complexity during learning.** A fundamental property of the theoretical model is that it provides an optimum performance bound to which the empirical behavior can be compared. To carry out this comparison, we plotted the value and complexity of each empirical trajectory against each other and compared them to the theoretically derived optimum (Fig 5). The green line represents the *value-complexity curve*, which is a theoretical bound on the maximal expected value (ordinate) that can be achieved for a given complexity (abscissa) level, as detailed subsection Computing the theoretical value-complexity curves of the Methods section. Initially, the empirical trajectories had low values but the value rapidly increased towards its maximal value of 0. Complexity, however, behaved differently for different mouse groups.

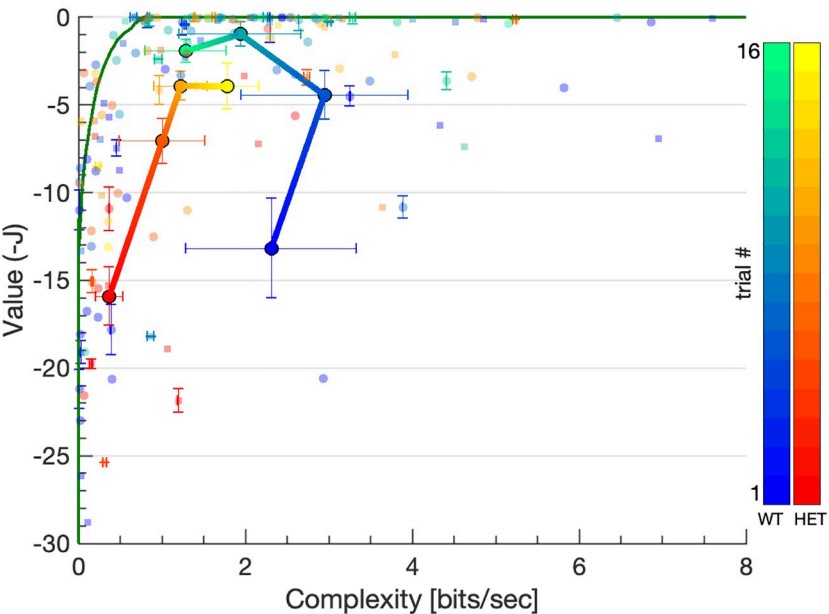

**Fig 5. Value-complexity curve.** Each point represents an empirical trajectory from a single release location (*N*). The axes show the value (ordinate) and complexity (abscissa) of each trajectory with the theoretically optimal curve plotted in green. Complexity tended to be lower for the mutant (heterozygous) animals compared to the wildtypes (warm and cool color scales respectively), and more so for females (circles) than for males (squares). While mean value tended to increased monotonically with training for both mutant and wildtype females (warm and cool gradient lines respectively), the mean complexity of wildtype females exhibited a non-monotonic profile, increasing on days 1-3 and decreasing on day 4. Trials from all six mouse batches are superimposed, with color hue indicating serial position within each batch. Large circles represent the daily mean value and complexity levels of wildtype (cool colors) and mutant (warm colors) female mice. Error bars are displayed for every 5th trial to reduce visual clutter.

For female wildtype mice, it exhibited a non-monotonic profile, increasing between days 1-2 and then decreasing, reaching the vicinity of the knee of the theoretical value-complexity curve on the late trials (cool gradient colored line). A two-sample Kolmogorov-Smirnov test showed a significant difference in trajectory complexity between days 2 and 4 for wildtype female mice ($D = 0.33$, $p = 0.03$). Thus, for these mice the learning dynamics can be partitioned into two phases: initial optimization (value increase) followed by late compression (complexity decrease). This two-stage learning dynamics was not observed in mutant females (warm gradient colored line) or male mice groups. As discussed below, this interaction between genotype and sex was significant.

We proceeded to quantitatively analyze the statistical properties of value and complexity as learning quantifiers over successive training days in the water maze. When compared on measures such as latency to platform, both wildtype mice and mice heterozygous to the mutated *Pogz* gene showed substantially equivalent rates of learning (Fig 6, left). A linear mixed effect model (fixed factors: day, sex and genotype; random factor: mouse) showed significant effects of day ($F(3, 595) = 12.1$, $p = 1.0 \times 10^{-7}$) and genotype ($F(1, 595) = 10.7$, $p = 0.0011$), with the heterozygous mice taking longer, on average, to reach the platform (see Fig 6, left). No significant effect of sex ($F(1, 595) = 0.201$, $p = 0.65$) or interactions with sex were observed (for example, the sex✕genotype interaction ($F(1, 595) = 1.47$, $p = 0.22$) was not significant).

The value tended to follow the latency to platform (Fig 6, center), except that value increased as latency to platform decreased. A linear mixed effect model (fixed factors: day, sex and genotype; random factor: mouse) showed significant effects of day ($F(3, 595) = 53.5$, $p = 1.3 \times 10^{-30}$) and genotype ($F(1, 595) = 4.81$, $p = 0.03$). There was also a weakly significant

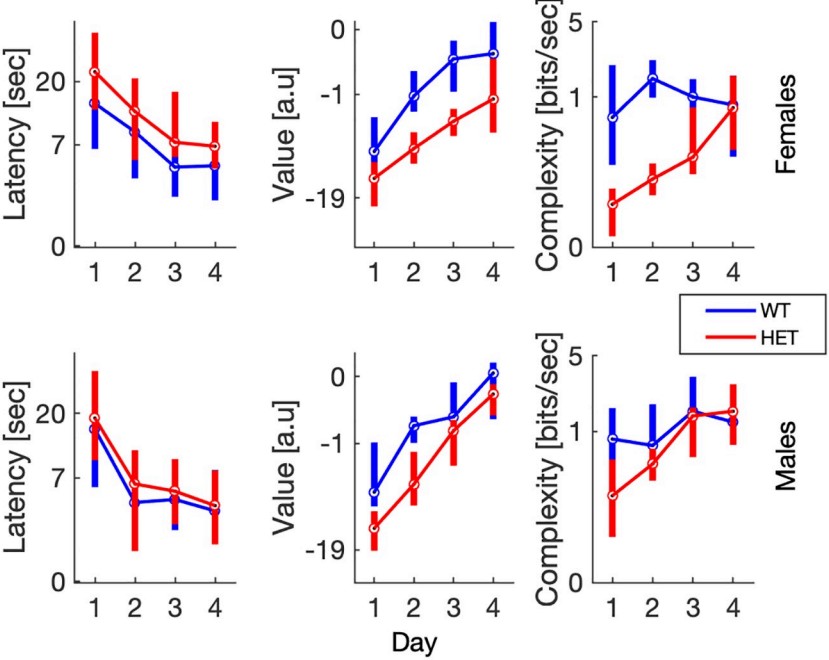

**Fig 6. Trajectory statistics.** Median path latency, value and complexity (ordinate) vs. trial day (abscissa) for female (top) and male (bottom) mice. Bottom and top bar edges indicate the 25th and 75th percentiles respectively. To reduce heteroscedasticity, ordinate data was transformed using a Box-Cox transform with power coefficients of: −0.29, 0.20, −0.19 (for latency, value and complexity data respectively).

genotype×day interaction ($F(3, 595) = 2.71$, $p = 0.044$). This reflected the somewhat faster increase in value of the wildtype (particularly of the female) relative to the heterozygous mice. As in the case of latency to platform, there were no significant effects or interactions with sex (for example, the sex×genotype interaction was not significant, $F(1, 595) = 0.291$, $p = 0.59$).

The new complexity measure introduced here showed sex × genotype interactions (Fig 6, left) and facilitated the discovery of interesting trajectory features. The linear mixed effect model (fixed factors: day, sex and genotype; random factor: mouse) showed significant effects of day ($F(3, 595) = 8.68$, $p = 1.2 \times 10^{-5}$) and genotype ($F(595, 1) = 20.3$, $p = 7.6 \times 10^{-6}$) as well as significant interactions for sex×day ($F(3, 595) = 4.87$, $p = 0.0023$), genotype×day ($F(3, 595) = 25.6$, $p = 1.2 \times 10^{-15}$) and sex×genotype×day ($F(3, 595) = 3.77$, $p = 0.011$). Indeed, complexity was substantially smaller for the female heterozygous mice relative to all other subgroups; i.e., the male heterozygous as well as the wildtype mice of both sexes, but mostly on days 1-3, reaching the level of wildtype mice on day 4.

We then examined the the swimming behavior of female heterzygous mice in order to find out why their complexity was reduced to such an extent. We observed that these mice had a tendency to practically stop moving and simply float for short periods of time mid-swim. To quantify this flotation behavior, we marked path segments in which the speed of the mouse was lower than 1/10 of its mean speed along the trajectory (Fig 7). Since such flotation behavior is consistent with the uncontrolled model, it reduced the integrated complexity along the swimming path. A linear mixed effect model for the number of floating episodes (fixed factors: day, sex and genotype; random factor: mouse) showed a significant genotype effect ($F(1, 606) = 33.1$, $p = 1.4 \times 10^{-8}$) as well as a sex×genotype interaction ($F(1, 606) = 9.81$, $p = 0.002$), confirming that the heterozygous females had a significantly larger number of such episodes relative to the other groups. These observations suggested that reduced complexity may be related,

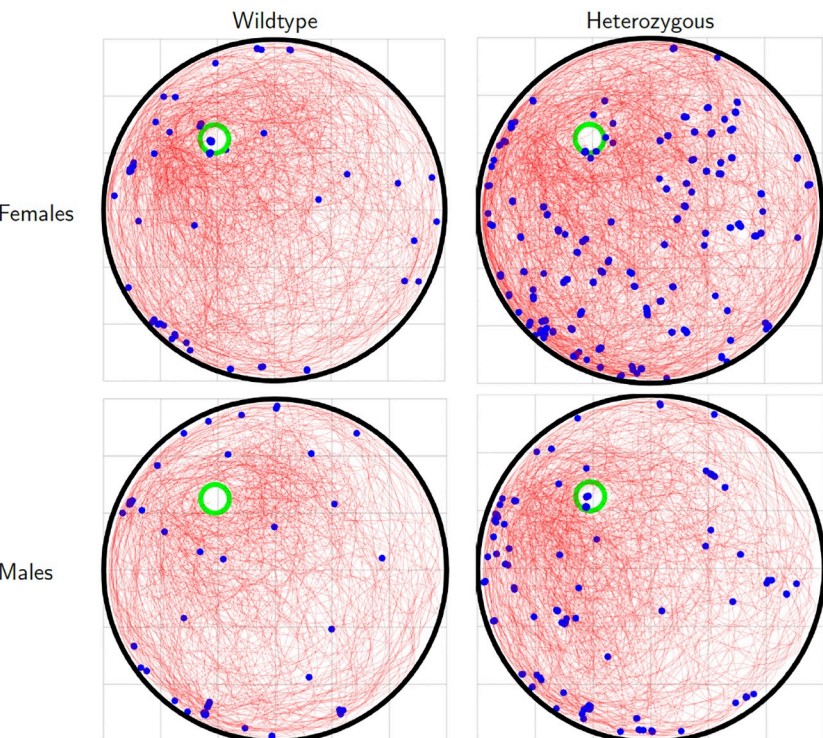

**Fig 7. Flotation behavior.** Trajectories of female (top) and male (bottom) wildtype (left) and heterozygous (right) mice released at the *NE* location. Blue circles indicate trajectory segments in which the speed of the mice was slower than 10% of the mean velocity along the trajectory.

at least partially, with slower swimming velocities. Indeed, a significant correlation was found between complexity and mean swimming velocities ($r(813) = 0.31$, $p = 5 \times 10^{-20}$) as well as a significant reduction in mean velocity between days 2 and 4 for trajectories of female wildtype mice (two sample Kolmogorov-Smirnov test, $d = 0.36$, $p = 0.016$).

## Discussion

### Summary

Navigational learning requires the determination of the forces needed to guide the movement of an object towards a desired location, typically under constraints such as minimizing latency or energy consumption. These problems have been studied by scientists and engineers for centuries. Major advances were made possible by the advent of the digital computer and the aerospace industry in the 1940s, leading to the development of feedback and optimal control theory [11], which are the pillars of modern navigation systems. While optimal and feedback control frameworks have been used to study sensorimotor systems [12–14], they have rarely been applied to mammalian navigational learning. This may be because control theory focuses on finding optimal trajectories by minimizing (or maximizing) a single performance criterion, whereas biological learning often requires satisfying several, possibly unknown and conflicting, optimization constraints.

We modeled mouse behavior in the water maze as a control system that operates optimally under complexity constraints. A control system consists of a dynamical system that can be steered using a control signal. Optimal control entails the selection of a control signal that optimizes a known value functional. Here, the dynamical system encapsulates the physical essence

of the problem—motion of the mouse through a viscous medium. The tendency of naive mice to swim in wide, quasi-circular arcs near the wall of the tank led us to model their trajectories with the dynamics of a stochastic, damped oscillator. Naive model trajectories were shaped by a balance between the tendency of the noise-free trajectories to converge spirally towards the center of the pool on the one hand, and the noise that drives the trajectories outwards on the other hand. This simple model captured properties of naive mice motion surprisingly well. The control signal consisted of the learned forces applied by the mouse to steer itself towards the platform. The complexity of the swimming paths was measured by how different they were from paths generated through a non-informative choice, in our case zero control, where swimming direction was determined by the dynamical system driven by isotropic Gaussian noise. Learning is quantified by the relaxation of the constraints on the complexity, making it possible to achieve higher value. We estimated the parameters of the problem (the dynamical system as well as the value functional) from data. This formulation makes it possible to define novel quantifiers of mouse behavior, namely value and complexity, which are theoretically-derived and uncover new features of the learning process.

We validated the model by using the initial and final trials of wildtype mice as training data for the uncontrolled and controlled model parameters respectively, and then used these parameters to estimate the value and complexity of the remaining trials of wildtype mice as well as all mutant mice trials. Thus, the final trajectories of wildtype mice, as well as those which were statistically similar to them, tended to cluster near the optimal value just below 0 in the value-complexity curve (Fig 5 top green horizontal bounding line). Similarly, all initial wildtype trajectories, and those similar to them, clustered near the minimal complexity of 0 (Fig 5 left green vertical bounding line). Other trajectories, whose properties diverged from both initial and final wildtype ones, were scattered over the value-complexity plane, with their distance from the ordinate and abscissa reflecting their divergence from optimal and naive behaviors respectively.

We illustrated the usefulness of this approach by comparing data from wildtype and mutant mice. The new quantifiers were more sensitive than the standard measures of mouse behavior (such as latency to platform) to differences in the behavior between mouse strains. They revealed behavioral features that were undetected by standard performance measures used to quantify behavior in the water maze.

Importantly, the current work was designed to provide a normative model of the trajectory learning process in the water maze using first principles such as Newtonian dynamics, optimal control theory and information theory. In consequence, the model deals with behavioral variables—the motion trajectories—and not with the underlying neural mechanisms. Nevertheless, the model provides information about high-order internal variables—the weighing matrices for the value and the value-complexity trade-off variable $\beta$, which can be used to link neural processes with the observed dynamics of learning. Importantly, $\beta$ provides an efficient summary statistic of the learning state of the animal at a given time. As mice gained more detailed information regarding the location of the platform, and found out how to couple this information with the appropriate motor commands, they were able to generate more precise movements towards the platform from any point in the tank. This process is quantified by the increase in the estimated values of $\beta$.

## Does it work?

The model is highly simplified in that the mouse is modelled as a point particle, and the introduction of the central force that imposed the tendency for circular swimming paths may seem artificial. Nevertheless, despite its simplicity, the model captures and quantifies subtle

trajectory features. First, the numerical values of the estimated parameters were reasonably close to the (very few) reported values in the literature [15, 16]. Second, we compared four properties of the measured swimming paths to those of paths generated by the model. For three of these (total path length, latency to platform, and mean distance to the platform during the swimming path), the model paths matched well with the observed ones. Mean velocity had the correct range of values, although it failed to show the predicted non-monotonic dependence on the trade-off parameter $\beta$. This may be due to the noise introduced by numerical differentiation, which required additional smoothing that is not part of the model.

What accounts for the effectiveness of the model? First, because of learning is measured by a single parameter, $\beta$, the model made it possible to evaluate the day to day changes in the control used by the mice directly from the observed data. Next, the quadratic value functional encapsulated well the time and energy costs inherent to the underlying biological mechanisms. Finally, the complexity constraint is theoretically grounded in large deviations theory [5]. The combination of all three provides a simple yet powerful model.

Obviously, the model can be refined. For example, a more realistic description of swimming trajectories could be obtained by using non-linear dynamics or a different noise model, e.g., multiplicative, control-dependent noise which has been proposed as more suitable for the description of sensorimotor behavior than additive, control-independent one [17]. The physical constraints could be captured in a more natural way by imposing a hard boundary corresponding to the walls of the water maze instead of the central force we used. One disadvantage of these approaches is that they may require the introduction of artificial devices to reproduce the tendency of mice to swim in circular arcs close to the walls on their initial exposure. More fundamentally however, we simplified these aspects of the model in order to connect a "microscopic", moment-by-moment description of mouse behavior with a single parameter that controls the "macroscopic" learning dynamics through a transparent, tractable formalism, allowing the estimation of parameters from empirical trajectory ensembles.

The model could also be extended by treating the state vectors of both mouse and platform as unknown variables which need to be estimated by the mouse. In its current formulation, the model implicitly assumes that the mouse knows its exact location and velocity. Real mice, however, have only imprecise knowledge of their location and velocity, and must therefore generate, and presumably update, internal estimates of these values. Similarly, the model assumes that trained mice know the exact, stationary, position of the platform. A more comprehensive model model could use noisy estimates instead of the (fully known) mouse and platform state variables. Indeed, an important component of control theory, which was not explicitly utilized in this work, deals with the problem of optimally estimating unknown states based on noisy observations. In the case of linear observations with Gaussian noise, the maximum likelihood estimator is the well known Kalman filter [18]. Importantly, it is mathematically equivalent, or dual, to the LQR problem, where the cost functional weight matrices $Q$ and $R$ are interpreted as the precision (inverse covariance) matrices of the prior state estimate and observation noise respectively. Furthermore, the optimal control of a linear Gaussian system with linear Gaussian observations is given by the same feedback gain as in Eq 9 but with the estimated state replacing the real one (a result known as the *certainty equivalence principle* [4]). Thus, replacing the mouse and platform states with noisy, linear observations would not alter the form of the optimal control solution. Rather, it would decrease the relative weight of the state term in the value functional. While beyond the scope of this work, such an extension of the model also suggests possible comparisons between the state estimation variables in the model and in the brain, as indexed by electrophysiological data from place cells in the hippocampus or grid cells in the entorhinal cortex. This extension would provide a full closed loop model relating neuronal activity to navigational learning and behavior.

## Is the model novel?

While previous attempts have been made to model rodent navigational learning, they have mostly focused on explaining spatial map formation in terms of hippocampal place cell connectivity or activation features [19–24]. The current model is different, in that it supplies a high-level description of the learning process itself, without linking it directly with its neural implementation. Furthermore, while previous models describe learning dynamics using an array of neural-network weight and activation parameters [19, 20] the current model uses a single parameter, namely $\beta$, for the same purpose (Fig 3). These differences distinguish the current model from previously suggested models of water maze navigational learning, making it difficult to directly compare them.

In recent years there has been increasing use of information theoretic measures in machine learning and neuroscience. In a series of studies, Frankland and coworkers [25–27] demonstrated how entropy and KL divergence may be used as sensitive quantifiers of water maze task performance. The spatial distribution of each path in these studies was approximated by a Gaussian distribution, and then various information theoretic measures were extracted from these distributions. While our model uses information theoretic measures, they are applied in different ways. For example, our model does not assume that the spatial distributions of the paths are Gaussian. Rather, the dynamical noise, consisting of the local discrepancies between the empirical behavior and the prediction of the model at each point along the path, is normally distributed.

The current model can be viewed as an analytically tractable formulation of the reinforcement learning framework [1] in the context of linear dynamical systems. Policies consist of selecting the appropriate control signals. Thus, our model falls within current frameworks for studying agents learning to operate in a known environment (e.g. [28]), but has the advantage of closed form solutions for the optimal policies using the Kalman gain (Eq 9).

The introduction of complexity constraints constitutes the most important theoretical contribution of the current paper to modeling behavior in the water maze, providing new insights into the learning process. The complexity cost is situated within a general theoretical framework relating path optimization and complexity constraints via the "free energy" functional [29]. Optimal and adaptive control, and in particular the LQR with Gaussian noise, were initially framed as entropy minimization problems by Saridis [30]. Later work by Todorov [31] and Kappen [32] showed that a family of non-linear, stochastic optimal control problems can be efficiently solved by writing the control cost as a KL divergence. Recently, a similar heuristic has been proposed as a basis for biologically plausible mechanisms underlying the brain's ability for flexible, yet biased, planning and decision making [33]. In contradistinction to these models, here we use the KL divergence, relative to a naive prior, as a quantifier for computational constraints on goal directed behavior, rather than a heuristic for simplification of certain non-linear optimal control problems.

The combination of value and costs within the free energy functional formalism (Eq 18) is related to rate distortion theory and the information-bottleneck method [34, 35]. In the information-bottleneck case, $\beta$ quantifies the mutual information between an internal variable (e.g., the compressed representation of relevant sensory information in the brain of the mouse) and a target variable (e.g., the distribution of optimal control vectors from each point in phase space). In contrast, here we do not have access to the joint distribution of sensory inputs and optimal actions. Thus, $\beta$ does not directly control mutual information between these variables and a compressed internal representation. Instead, $\beta$ controls the tradeoff between policy complexity and the LQR value. Complexity can nevertheless be considered as a proxy for compression, where maximal compression ($\beta = 0$) corresponds to the

behavior of naive animals while full information ($\beta = \infty$) corresponds to the optimal LQR solution.

## Was it worth the effort?

Of the two quantifiers we used here, the value and the complexity, the value functional is closer to standard measures used to quantify mouse behavior in the water maze, such as latency to platform. While the choice of a suitable quantifier remains somewhat arbitrary (see [26] for a comparison between popular performance measures), the value as defined here is a theoretically-derived optimal choice, in the sense that it estimates the animal's own performance criterion (at least on average across mice). Furthermore, we show here that the value functional is more informative than the latency to platform. For example, while the latency to platform reached saturation by the third day of training and potentially even before, the value continued to increase monotonically throughout training (Fig 6).

The most important results of this study involve the other quantifier we introduced in this work, the complexity of the swimming paths. First, we observe that at least in wildtype female mice, complexity exhibited non-monotonic behavior during training in that it first increased and then decreased (Figs 5 and 6). This observation suggests that the learning process in the water maze can be roughly divided into two consecutive stages: *path optimization* and *path simplification*. In the first stage, task performance was optimized (increasing value), while behavior became more complex (increasing complexity). In the second stage, complexity showed a downward trend, representing simplification of the swimming paths. In the mutant mice, this behavior was not observed, and complexity increased throughout learning, together with value.

Interestingly, a similar dual-stage learning process has recently been observed in deep neural network learning dynamics [36], where the learning process has also been shown to consist of two stages also: prediction optimization, corresponding to value increase in our setting, followed by data compression, corresponding to complexity reduction. This similarity may reflect a fundamental feature of learning dynamics in general, suggesting that initially, high complexity levels may be utilized to optimize performance (value), whereas at later stages of learning irrelevant complexity is discarded to obtain simpler solutions while not compromising the performance.

The other important result of this paper consists of the use of complexity to differentiate between the behavior of WT and mutant mice (Figs 6 and 7). In the mutant mice, particularly in females, complexity was overall lower than in WT mice. The difference between males and females resulted in an interaction between genetic status and sex. This interaction was not observed in the latency to platform, and would have been missed using standard measures of behavior in the water maze. We therefore looked specifically for those features of the swimming paths that could cause this reduction of complexity in the female mutant mice. We found periods of almost motionless floating that were more common in female, mutant mice than female wildtype mice or male mice of both genetic types. These periods reduced total path complexity since motionless periods were more consistent with the uncontrolled than with the controlled model. While they did somewhat increase latency to platform (Fig 6), this increase was hardly detectable given the overall variability in the data. In contrast, these episodes affected the complexity very strongly. Complexity served here as a powerful tool for identifying novel behavioral features that differentiate between mice of different genotypes and sex. In particular, the reduced complexity of the mutant mice is consistent with low IQ and abnormal behavior observed in humans with mutations in *POGZ*, although in humans an interaction with gender has not been described.

## Methods

### Ethics statement

All experiments were approved by the Institutional Animal Care and Use Committee. The Hebrew University is an AAALAC accredited institution.

### Experimental procedures

For a detailed description of the water maze spatial learning task protocol see [37]. We analyzed data from wildtype mice and mice with a heterozygous mutation in the *Pogz* gene (pogo transposable element-derived protein with zinc finger domain). The generation of the *Pogz*$^{+/-}$ mice with deletion of exons 13-19 has been described previously [38]. Heterozygous loss-of-function mutations in the human *POGZ* gene are associated with intellectual disability and autism spectrum disorder independent of gender [39]. The heterozygous progeny was generated by crossing heterozygous mice with wildtype mice. All mice had a C57BL/6 genetic background. Both male and female animals, mutants and their wildtype littermates were used for the behavioral experiments.

For analysis and parameter estimation we used a data-set of water maze trajectories from $M = 51$ mice (WTs: 11 males, 13 females; HETs: 12 males, 15 females). The full data set thus consisted of $51 \times 4 \times 4 = 816$ trials, 49 of which were excluded from analysis due to missing samples or measurement errors, resulting in a total of 767 analyzed trials.

### Model discretization

To compute the discrete-time matrices (Eqs 10, 11 and 13), we introduce the matrix exponential operator which is defined, for any square matrix $M$, by:

$$\exp M = \sum_{n=0}^{\infty} \frac{M^n}{n!}. \tag{26}$$

The discrete-time approximations of $A$ and $B$ can now be defined as follows:

$$A_{\Delta t} = \exp(A\Delta t), \tag{27}$$

and:

$$B_{\Delta t} = A^{-1}(A_{\Delta t} - I)B. \tag{28}$$

The discrete-time approximation of the noise covariance matrix, $\Sigma_\xi$, is denoted by $\Sigma_{\Delta t}$, and given by the solution of the following Lyapunov equation:

$$A\Sigma_{\Delta t} + \Sigma_{\Delta t}A^T - A_{\Delta t}\Sigma A_{\Delta t}^T + \Sigma_\xi = 0, \tag{29}$$

which can be efficiently computed; e.g., using the MATLAB built-in `lyap` function. Finally, the discrete-time approximation of the cost functional weight matrices, $Q$, $R$ and $N$, denoted by $Q_{\Delta t}$, $R_{\Delta t}$ and $N_{\Delta t}$ respectively, can obtained via the following relations [40]:

$$\begin{pmatrix} Q_{\Delta t} & N_{\Delta t} \\ N_{\Delta t}^T & R_{\Delta t} \end{pmatrix} = \Phi_{22}^T \Phi_{12}, \tag{30}$$

with:

$$\exp\left(\begin{pmatrix} -A^T & 0 & Q & N \\ -B^T & 0 & N^T & R \\ 0 & 0 & A & B \\ 0 & 0 & 0 & 0 \end{pmatrix}^T \Delta t\right) = \begin{pmatrix} \Phi_{11} & \Phi_{12} \\ 0 & \Phi_{22} \end{pmatrix}. \tag{31}$$

## Estimating model parameters

We use the trajectories of wildtype mice to estimate the most likely model parameter values given the empirical data. We start by computing the log-likelihood of the free model by considering the residual terms:

$$\epsilon_t^i = \mathbf{x}_{t+1}^i - A_{\Delta t}\mathbf{x}_t^i \tag{32}$$

which, under the free model assumptions, should be independent, zero mean Gaussian random variables with covariance matrix $\Sigma_{\Delta t}$:

$$P(\epsilon_t^i) \sim \mathcal{N}(0, \Sigma_{\Delta t}). \tag{33}$$

Now, we can express the free model log-likelihood:

$$L(\theta^0 \mid \{\mathbf{x}_1^i \ldots \mathbf{x}_{T_i}^i\}_{i=1}^{M_{WT}}) = \log\prod_{i=1}^{M_{WT}}\prod_{t=1}^{T_i-1}P(\mathbf{x}_{t+1}^i \mid \mathbf{x}_t^i) = \log\prod_{i=1}^{M_{WT}}\prod_{t=1}^{T_i-1}\frac{e^{-\frac{1}{2}\epsilon_t^{i^T}\Sigma_{\Delta t}^{-1}\epsilon_t^i}}{\sqrt{(2\pi)^4\det\Sigma_{\Delta t}}} =$$

$$-\sum_{i=1}^{M_{WT}}\frac{T_i-1}{2}\left(\frac{1}{T_i-1}\sum_{t=1}^{T_i-1}\epsilon_t^{i^T}\Sigma_{\Delta t}^{-1}\epsilon_t^i + \log\det\Sigma_{\Delta t} + 4\log 2\pi\right) \tag{34}$$

where $\theta^0 = (k, \gamma, \sigma_q, \sigma_p)$ are the free model parameters and $\epsilon_t^i$ are the discretized free model residuals (Eq 32) using the sampled trajectory points $\mathbf{x}_1^i, \ldots, \mathbf{x}_{T_i}^i$ for the first trajectories of the $i$-th wildtype mouse ($i = 1, \ldots, M_{WT}$). Numerically maximizing the log-likelihood function over the training data yields the maximum likelihood estimate of the free model parameters:

$$\theta_{ML}^0 = (\hat{k}, \hat{\gamma}, \hat{\sigma}_q, \hat{\sigma}_p) = \arg\max_{\theta^0} L(\theta^0 \mid \{\mathbf{x}_1^i, \ldots, \mathbf{x}_{T_i}^i\}_{i=1}^{M_{WT}}). \tag{35}$$

The estimated continuous time dynamics matrix and noise covariance were:

$$\hat{A} = \begin{pmatrix} 0 & 0 & 1 & 0 \\ 0 & 0 & 0 & 1 \\ -0.18 & 0 & -0.02 & 0 \\ 0 & -0.18 & 0 & -0.02 \end{pmatrix}, \tag{36}$$

and

$$\hat{\Sigma}_{\xi} = \begin{pmatrix} 1.09 & 0 & 0 & 0 \\ 0 & 1.09 & 0 & 0 \\ 0 & 0 & 19.8 & 0 \\ 0 & 0 & 0 & 19.8 \end{pmatrix}. \tag{37}$$

Next, we use the last trial trajectories of each wildtype mouse to estimate the matrices that define the cost functional, $Q$, $R$ and $N$. Estimating a cost functional from a set of trajectories is known as the *inverse optimal control problem*, and it goes back at least to the early 1960s [41]. In this problem, rather than starting with a known optimization functional and finding the optimal trajectories, the optimized trajectories are known and we want to find a quadratic cost functional that can explain them. The inverse optimal control problem is ill-defined since typically there are many weight matrices that result in the same steady state feedback gain and therefore in the same optimal trajectory. Thus, in order to obtain a unique correspondence between the steady state optimal feedback gain $K$ and the weight matrices which produce it, we constrain the solution to weight matrices of the following form (see [42] for details):

$$Q = K^T K, \quad R = \begin{pmatrix} 1 & 0 \\ 0 & 1 \end{pmatrix}, \quad N = -K^T. \tag{38}$$

For this choice of parameters, the functional $J$ (Eq 7) reduces to

$$\mathbb{E}\left[\frac{1}{2}\int_0^T \left(\|K((\mathbf{x}(t) - \bar{\mathbf{x}}))\|^2 + \|\mathbf{u}(t)\|^2 - 2K(\mathbf{x}(t) - \bar{\mathbf{x}})^T \mathbf{u}(t))\right)dt\right] \tag{39}$$

so that $J = 0$ identically for the optimal solution. As shown in the Boundary conditions and transients section, this choice of parameters also simplifies the solution of the optimal control problem by eliminating temporal transients.

We proceed to estimate the optimal feedback gain matrix, $K$, which best fits the late trajectory dynamics. We use maximum likelihood on the optimal control model (Eqs 5–9) with the 2X4 entries of $K$ as the unknown parameters. To obtain a likelihood function similar to Eq 34 we express the discretized optimal control model residuals as:

$$\epsilon_t^{i*} = \mathbf{x}_{t+1}^i - [A_{\Delta t}\mathbf{x}_t^i + B_{\Delta t}K_{\Delta t}(\mathbf{x}_t - \bar{\mathbf{x}})] \tag{40}$$

which under the noise assumptions, are independent, zero mean Gaussian variables with covariance matrix $\Sigma_{\Delta t}$ (Eq 29):

$$P(\epsilon_t^{i*}) \sim \mathcal{N}(0, \Sigma_{\Delta t}). \tag{41}$$

Letting $\tilde{\mathbf{x}}_1^i, \ldots, \tilde{\mathbf{x}}_{T_i}^i$ denote the *last* trajectory taken by the $i$-th wildtype animal ($i = 1, \ldots, M_{WT}$), we can write the log-likelihood as:

$$L \quad (\theta^* \mid \{\tilde{\mathbf{x}}_1^i \ldots \tilde{\mathbf{x}}_{T_i}^i\}_{i=1}^{M_{WT}}) =$$

$$-\sum_{i=1}^{M_{WT}} \frac{T_i - 1}{2}\left(\frac{1}{T_i - 1}\sum_{t=1}^{T_i-1} \epsilon_t^{i*T} \Sigma_{\Delta t}\epsilon_t^{i*} + \log \det\Sigma_{\Delta t} + 4\log 2\pi\right) \tag{42}$$

where $\theta^* = K$ is the optimal control feedback gain matrix. All other variables in Eq 42 can be

computed using the known matrices $A$, $B$, $\Sigma$. The estimated optimal feedback gain matrix is obtained by maximizing Eq 42:

$$\theta^*_{ML} = \hat{K} = \arg\max_{\theta^*} L(\theta^* \mid \{\tilde{\mathbf{x}}^i_1, \ldots, \tilde{\mathbf{x}}^i_{T_i}\}^{M_{WT}}_{i=1})). \tag{43}$$

Since $K$ is a 2X4 matrix it has 8 parameters which need to be estimated. In practice, however, it can be well approximated by a matrix with the following structure:

$$\theta^* = K = \begin{pmatrix} K_r & 0 & K_t \cos(K_\alpha) & -K_t \sin(K_\alpha) \\ 0 & K_r & K_t \sin(K_\alpha) & K_t \cos(K_\alpha) \end{pmatrix} \tag{44}$$

in which the parameter $K_r$ describes a restoring force proportional to the displacement from the platform, whereas the two remaining parameters, $K_t$ and $K_\alpha$, describe a rotation of the velocity vector that tends to point it in the direction of the platform. Thus the radial component $K_r$ can be thought of as a force by which the animal attempts to reduce its distance to the platform, while $K_t$ and $K_\alpha$ represent the animal's effort to rotate itself towards the correct azimuth.

Using the maximum likelihood estimated parametric form of $\hat{K}$ (Eq 25), the values obtained for $Q$ and $N$ are:

$$\hat{Q} = \hat{K}^T \hat{K} = \begin{pmatrix} 0.40 & 0 & 0.48 & -0.58 \\ 0 & 0.40 & 0.3 & 0.4 \\ 0.48 & 0.58 & 1.4 & 0 \\ -0.58 & 0.48 & 0 & 1.4 \end{pmatrix} \times 10^{-1}, \tag{45}$$

and

$$\hat{N} = -\hat{K}^T = \begin{pmatrix} -0.20 & 0 \\ 0 & -0.20 \\ -0.24 & -0.29 \\ 0.29 & -0.24 \end{pmatrix}. \tag{46}$$

## Computing the theoretical value-complexity curves

Once we estimated the free and controlled model parameters, using the initial and final trials respectively (Eqs 34 and 42), we can calculate the optimal trade-off between value and complexity for each value of $\beta$ in the free energy functional (Eq 18). For this we derive a closed form solution of the free energy minimization problem. We need to determine the complexity constrained optimal control signal, $\mathbf{u}^\beta_t$ which minimizes the free energy (Eq 18) at any given $\beta$ for the linear model dynamics. Since both $P_0$ and $P_\beta$ are normal distributions, we can calculate

the complexity cost explicitly using the formula for the KL divergence between two Gaussians:

$$
\begin{aligned}
I_{\Delta t}[\mathbf{x}_t, \mathbf{u}_t] \quad &= \frac{1}{2}\sum_{t=1}^{T-1} D_{KL}\big(P_\beta(\mathbf{x}_{t+1} \mid \mathbf{x}_t)\|P_0(\mathbf{x}_{t+1} \mid \mathbf{x}_t)\big) = \\
&\frac{1}{2}\Big((\mu_\beta - \mu_0)^T \Sigma_0^{-1}(\mu_\beta - \mu_0) + \mathrm{Tr}(\Sigma_0^{-1}\Sigma_\beta) - \log\frac{\det \Sigma_0}{\det \Sigma_\beta} - 4\Big)
\end{aligned}
\tag{47}
$$

where $\mu_\beta$, $\Sigma_\beta$ and $\mu_0$, $\Sigma_0$ denote the means and covariances of $P_\beta$ and $P_0$ respectively. Since $\Sigma_0 = \Sigma_\beta = \Sigma_{\Delta t}$, the last three terms in Eq 47 cancel out and the complexity cost reduces to the mean difference term:

$$
I_{\Delta t}[\mathbf{x}_t, \mathbf{u}_t] = \frac{1}{2}(\mu_\beta - \mu_0)^T \Sigma_0^{-1}(\mu_\beta - \mu_0) = \frac{1}{2}\mathbf{u}_t^T B_{\Delta t}^T \Sigma_{\Delta t}^{-1} B_{\Delta t}\mathbf{u}_t.
\tag{48}
$$

The mean free energy (Eq 18) can thus be rewritten as:

$$
\begin{aligned}
F_{\Delta t}[\mathbf{x}, \mathbf{u}, \beta] &= \frac{1}{2}\sum_{t=1}^{T}\Big(\mathbf{u}_t^T B_{\Delta t}^T \Sigma_{\Delta t}^{-1} B_{\Delta t}\mathbf{u}_t + \\
&\beta\big((\mathbf{x}_t - \bar{\mathbf{x}})^T Q_{\Delta t}(\mathbf{x}_t - \bar{\mathbf{x}}) + \mathbf{u}_t^T R_{\Delta t}\mathbf{u}_t + 2(\mathbf{x}_t - \bar{\mathbf{x}})^T N_{\Delta t}\mathbf{u}_t\big)\Big) = \\
&\frac{\beta}{2}\sum_{t=1}^{T}\Big((\mathbf{x}_t - \bar{\mathbf{x}})^T Q_{\Delta t}(\mathbf{x}_t - \bar{\mathbf{x}}) + \mathbf{u}_t^T R_{\Delta t}^\beta \mathbf{u}_t + 2(\mathbf{x}_t - \bar{\mathbf{x}})^T N_{\Delta t}\mathbf{u}_t\Big),
\end{aligned}
$$

where we denote:

$$
R_{\Delta t}^\beta = R_{\Delta t} + \frac{B_{\Delta t}^T \Sigma_{\Delta t}^{-1} B_{\Delta t}}{\beta}.
\tag{49}
$$

Thus we can restate the complexity constrained LQR problem as a standard LQR problem with a $\beta$-regularized control cost weight matrix $R_{\Delta t}^\beta$ replacing $R_{\Delta t}$. The optimal complexity constrained control signal at each $\beta$ is given by:

$$
\mathbf{u}_t^\beta = -K_{\Delta t}^\beta(\mathbf{x}_t - \bar{\mathbf{x}})
\tag{50}
$$

where $K_{\Delta t}^\beta$ is computed from the discrete dynamics and cost functional matrices (see Computing the optimal feedback gain below for details).

We can now use $\mathbf{u}_t^\beta$, the mean complexity constrained optimal control signal at each $\beta$ value, to compare the value-complexity trade-off of the empirical trajectories with the theoretical optimum. To do so, we simulated the optimal control dynamics at each $\beta$ and each release point using the solution of the discrete-time problem (Eq 10), with the maximum likelihood estimates for $A_{\Delta t}$ and $\Sigma_{\Delta t}$ (Eq 35) for the free dynamics parameters and the theoretically computed $\mathbf{u}_t^\beta$ (Eq 58) for the optimal control signal. The simulations were computed at 50 logarithmically spaced $\beta$ values between $\beta = 10^{-5}$ and $\beta = 10^5$ and the value and complexity measures (Eqs 11 and 14) were averaged over 1,000 repetitions of the simulation with identical initial conditions. Since many of the experimental paths were missing their first few seconds due to experimental limitations, we replaced the nominal release point with the mean *empirical* starting point; i.e., the first position registered by the tracking device, over all trials from a given release point. This resulted in the *value-complexity curve* (Fig 5) for each of the four (mean) release positions. For each complexity level (abscissa) the value-complexity curve shows the maximal value (ordinate) which can be obtained by a trajectory with that complexity level,

starting at the mean release position. Equivalently, for each value (ordinate), the curve marks the minimal amount of complexity (abscissa) required to achieve it.

## Computing the optimal feedback gain

The continuous-time optimal control feedback gain matrix, $K$, is computed from the continuous-time dynamics and value functional matrices as follows. Generally, $K = K(t)$ is a time-varying gain:

$$K(t) = -R^{-1}(B^T S(t) + N^T) \tag{51}$$

where $S(t)$ is the solution of the following *differential Riccati equation*, see Derivation of the Riccati equation for details:

$$-\dot{S}(t) = A^T S(t) + S(t)A - (S(t)B + N)R^{-1}(B^T S(t) + N^T) + Q. \tag{52}$$

In the case $\beta = \infty$ (optimal control) the situation was simplified substantially, since the matrices solving the inverse control problem were selected so that $S(t) = 0$ is the solution to Eq 52, so that $K(t)$ is constant (see the Boundary conditions and transients section below). For finite $\beta$, we observed that $S(t)$ rapidly converged to a steady-state value. We neglected the effects of the rapidly decaying transients by using the solution of the following quadratic matrix equation, known as the continuous-time *algebraic Riccati equation* [4]:

$$A^T S + SA - (SB + N)R^{-1}(B^T S + N^T) + Q = 0. \tag{53}$$

The resulting value of $S$ was used to compute the feedback gain matrix $K$ using Eq 51.

In the discrete-time case, the optimal control feedback gain matrix, $K_{\Delta t}$, is given by:

$$K_{\Delta t} = (B_{\Delta t}^T S_{\Delta t} B_{\Delta t} + R_{\Delta t})^{-1}(B_{\Delta t}^T S_{\Delta t} A_{\Delta t} + N_{\Delta t}^T), \tag{54}$$

where $S_{\Delta t}$ is the solution of the following *discrete-time algebraic Riccati equation* [4]:

$$S_{\Delta t} = A_{\Delta t}^T S_{\Delta t} A_{\Delta t} - (A_{\Delta t}^T S_{\Delta t} B_{\Delta t} + N_{\Delta t})(B_{\Delta t}^T S_{\Delta t} B_{\Delta t} + R_{\Delta t})^{-1}(B_{\Delta t}^T S_{\Delta t} A_{\Delta t} + N_{\Delta t}^T) \\ + Q_{\Delta t}. \tag{55}$$

Section Computing the theoretical value-complexity curves, shows how to reduce the complexity-constrained optimal control to a discrete LQR problem with a modified cost functional. The feedback gain matrix, $K_{\Delta t}^\beta$, can then by expressed using a formula analogous to Eq 54:

$$K_{\Delta t}^\beta = (B_{\Delta t}^T S_{\Delta t}^\beta B_{\Delta t} + R_{\Delta t}^\beta)^{-1}(B_{\Delta t}^T S_{\Delta t}^\beta A_{\Delta t} + N_{\Delta t}^T), \tag{56}$$

where:

$$R_{\Delta t}^\beta = R_{\Delta t} + \frac{B_{\Delta t}^T \Sigma_{\Delta t}^{-1} B_{\Delta t}}{\beta}, \tag{57}$$

and $S_{\Delta t}^\beta$ is the solution of the following discrete-time algebraic Riccati equation:

$$S_{\Delta t}^\beta = A_{\Delta t}^T S_{\Delta t}^\beta A_{\Delta t} - (A_{\Delta t}^T S_{\Delta t}^\beta B_{\Delta t} + N_{\Delta t})(B_{\Delta t}^T S_{\Delta t}^\beta B_{\Delta t} + R_{\Delta t}^\beta)^{-1}(B_{\Delta t}^T S_{\Delta t}^\beta A_{\Delta t} + N_{\Delta t}^T) \\ + Q_{\Delta t}, \tag{58}$$

which is analogous to Eq 55 in the standard LQR case. Thus, the free energy minimization problem can be reduced to a standard LQR problem and solved using the same methods [35].

## Estimating $\beta$

To compare the value and complexity of the empirical trials to the theoretical optimum we need to estimate $\beta$ for the empirical trials. We do so by using maximum likelihood again, as in Eq 42, with the trajectories of each mouse taken as observations and $\beta$ as the estimated parameter. The estimated $\beta$ is then used in Eq 50 to determine the control signal.

Although we considered $\beta$ to be a parameter characterizing the learning stage, rather than the specific swimming path of the mouse, the estimates of $\beta$ turned out to be sensitive to the trial-specific starting state. We therefore estimated a single $\beta$ value for each training day and each mouse by grouping the trajectories from all four starting locations for each mouse/day combination. We expressed $\log(\beta)$ as a quadratic function of the day:

$$\log(\beta_i)(day) = b_0 + b_1(day) + b_2(day)^2, \tag{59}$$

where $i = 1, \ldots, M$ and the parameters $b_0$, $b_1$ and $b_2$ are estimated for each mouse separately. We used a quadratic function since it is the simplest one that can account for the non-linear dependence of $\beta$ on training day, which was observed in many cases.

Given the values of $\beta$ for each mouse on each of the four training days, we can calculate the value as well as the complexity of all the empirical trajectories. Note that the value-complexity curve is an expectation, and therefore does not bound the single path values. Nevertheless, we do not expect single path values far beyond the average curve.

## Large deviations theory and Sanov's theorem

In this section we provide a theoretical justification for the choice of our complexity functional (Eq 14) based on a result from large deviations theory known as Sanov's theorem [5]. The theory of large deviations is concerned with the asymptotic behavior of extreme values, i.e., far from the expected ones, of sequences of probability measures. As an example consider the following question: what is the probability that $\frac{1}{n}\sum_i X_i$ is larger than $\frac{3}{4}$ when $X_i$ are all drawn i.i.d. from a $Bernoulli\left(\frac{1}{3}\right)$ distribution? This event represents a large deviation from the expected value of $\frac{1}{3}$, and its probability decays exponentially with $n$, i.e., it is equal or smaller to $e^{-n\alpha}$ for some $\alpha$. The smallest such $\alpha$ (if it exists), giving the tightest bound on the probability, is an indication of how extreme is the large deviation.

The probability of such large deviations and their rate of decrease ($\alpha$ above) can be estimated using the following result, known as Sanov's theorem: let $X_1, X_2, \ldots X_n$ be i.i.d random variables with common distribution $Q$, and let $E$ denote an arbitrary set of probability distributions (which typically does not contain $Q$). Consider now the probability that the empirical distribution of the $X_i$'s belongs to the set $E$, and denote this probability as $Q^n(E)$. Sanov's theorem states that if $E$ fulfills a technical condition (it is equal to the closure of its interior) then:

$$\lim_{n\to\infty} \frac{1}{n}\log Q^n(E) = -D_{KL}(P^*||Q), \tag{60}$$

where,

$$P^* = \arg\min_{P\in E} D_{KL}(P||Q) \tag{61}$$

is the *information projection* of $Q$ onto $E$, i.e., the distribution in $E$ which is closest to $Q$ in the Kullback-Leibler (KL) divergence sense. In words, the exponential rate of decrease of the probability of drawing an atypical distribution is the KL divergence between the true distribution and the atypical one (or more generally, the information projection onto the set of atypical distributions).

This mathematical result implies that the difficulty in distinguishing between a typical and atypical distribution, using some statistical test, is determined by the KL divergence between them. Thus, the KL divergence between two distributions measures how unlikely it is for a sample drawn from one distribution to be mistakenly classified as originating from the other. In the context of our model, the complexity of a controlled trajectory is considered to be higher when it is less likely to be generated by naive mice. Letting $E$ denote the distribution of trajectories generated by a control signal achieving a certain value, Sanov's theorem implies that the likelihood for a such a trajectory to be generated by a naive mouse is determined by the KL divergence between the controlled and non-controlled trajectory distributions. This is precisely how our complexity measure (Eq 14) is defined.

## Derivation of the Riccati equation

In this section we show how to reduce the Linear Quadratic Regulater (LQR) optimization problem to that of solving the Riccati differential equation (Eq 52). Since this material is standard [4], we describe here only the case of continuous-time, deterministic systems. The discrete-time and the stochastic cases can be treated similarly (see [4] for details).

The LQR problem consists of finding a control signal which minimizes a quadratic cost functional subject to dynamics which are linear in the state and the control. The (deterministic) dynamics are given by (cf. Eq 5):

$$\dot{\mathbf{x}}(t) = A\mathbf{x}(t) + B\mathbf{u}(t), \tag{62}$$

and the quadratic cost functional can be written, in the general case, as follows (cf. Eq 7):

$$J \quad [\mathbf{x}(t), \mathbf{u}(t)] = \frac{1}{2}(\mathbf{x}(T) - \bar{\mathbf{x}})^T S_T (\mathbf{x}(T) - \bar{\mathbf{x}}) + \frac{1}{2}\int_0^T \left((\mathbf{x}(t) - \bar{\mathbf{x}})^T Q(\mathbf{x}(t) - \bar{\mathbf{x}}) + \right.$$
$$\mathbf{u}(t)^T R\mathbf{u}(t) + 2(\mathbf{x}(t) - \bar{\mathbf{x}})^T N\mathbf{u}(t))dt, \tag{63}$$

where $S_T$ is a positive semi-definite matrix weighing the cost of deviating from the the desired state, $\bar{\mathbf{x}}$, at the terminal time, $t = T$ (in our model, there is no terminal cost term, i.e., $S_T = 0$, and see also the following subsection). Such optimal control problems are solved using standard variational techniques, which result in a differential functional equation called (in this case) the Hamilton-Jacobi-Bellman (HJB) equation (or simply the Bellman equation in the discrete-time case). The HJB equation provides necessary and sufficient conditions for the optimality of a control signal with respect to a given cost functional. These conditions can be stated in terms of a set of differential equations involving the following Hamiltonian:

$$H[\mathbf{x}(t), \mathbf{u}(t), \lambda(t)] = \frac{1}{2}\left((\mathbf{x}(t) - \bar{\mathbf{x}})^T Q(\mathbf{x}(t) - \bar{\mathbf{x}}) + \right.$$
$$\mathbf{u}(t)^T R\mathbf{u}(t) + 2(\mathbf{x}(t) - \bar{\mathbf{x}})^T N\mathbf{u}(t)\right) + \lambda^T(t)\left(A\mathbf{x}(t) + B\mathbf{u}(t)\right), \tag{64}$$

where $\lambda(t)$ are Lagrange multipliers, also referred to in this context as the *co-state* coordinates. Using this formulation, the state and co-state dynamics can be expressed simultaneously via the following Hamiltonian equations:

$$\dot{\mathbf{x}}(t) = \frac{\partial H[\mathbf{x}(t), \mathbf{u}(t), \lambda(t)]}{\partial \lambda(t)}, \tag{65}$$

with the initial condition:

$$\mathbf{x}(0) = \mathbf{x}_0, \tag{66}$$

where $\mathbf{x}_0$ is the initial state of the system, and:

$$\dot{\lambda}^T(t) = -\frac{\partial H[\mathbf{x}(t), \mathbf{u}(t), \lambda(t)]}{\partial \mathbf{x}(t)}, \tag{67}$$

with the terminal condition:

$$\lambda(T) = S_T \mathbf{x}(T), \tag{68}$$

where $S_T$ is the terminal cost weight matrix defined in Eq 63 above. The condition for optimality of the control is given by:

$$\frac{\partial H[\mathbf{x}(t), \mathbf{u}(t), \lambda(t)]}{\partial \mathbf{u}(t)} = 0. \tag{69}$$

Performing the differentiations in Eqs 69 and 67 yields the following:

$$\mathbf{u}(t) = -R^{-1}(B^T\lambda(t) + N^T\mathbf{x}(t)), \tag{70}$$

and:

$$\dot{\lambda}(t) = -Q\mathbf{x}(t) - N\mathbf{u}(t) - A^T\lambda(t). \tag{71}$$

The co-state dynamics (Eq 71) can be solved via the ansatz:

$$\lambda(t) = S(t)\mathbf{x}(t). \tag{72}$$

Substituting Eq 72 into Eqs 70 and 71 gives:

$$\mathbf{u}(t) = -R^{-1}(B^T S(t) + N^T)\mathbf{x}(t), \tag{73}$$

and:

$$S(t)\dot{\mathbf{x}}(t) + \dot{S}(t)\mathbf{x}(t) = -Q\mathbf{x}(t) - N\mathbf{u}(t) - A^T S(t)\mathbf{x}(t), \tag{74}$$

which together with the state dynamics (Eq 65) yields the following equation:

$$-\dot{S}(t) = A^T S(t) + S(t)A - (S(t)B + N)R^{-1}(B^T S(t) + N^T) + Q. \tag{75}$$

Eq 75 is the Riccati differential equation (cf. Eq 52) which can be solved numerically by integrating backwards in time, starting from the terminal condition $S(T) = S_T$.

## Boundary conditions and transients

In this section we justify the use of the algebraic Riccati equation (Eq 53) instead of the differential one (Eq 52) for solving the optimal control problem in the watermaze model. The optimal control of a time-constrained, or *finite-horizon*, LQR problem typically contains a transient component ($S(t)$ in Eq 73) due to the co-state terminal condition ($S_T$ in Eq 63). In our model however, the cost functional (Eq 7) does not contain a terminal cost term, since that there is no additional penalty for failing to reach the platform at the end of the trial. This means that the transient term in the optimal control is zero at the terminal time, i.e., $S(T) = 0$. To show that the transient term is in fact zero always, we recall our choice of parameterization for the cost functional matrices $Q$, $R$ and $N$ in the inverse optimal control problem (see

Estimating model parameters section, Eq 38), namely:

$$Q = K^T K, \quad R = \begin{pmatrix} 1 & 0 \\ 0 & 1 \end{pmatrix}, \quad N = -K^T. \tag{76}$$

For this choice of parameters, the Riccati differential equation (Eq 75), reduces to the following form:

$$-\dot{S}(t) = A^T S(t) + S(t)A - S(t)BB^T S(t) - S(t)BN^T - NB^T S(t), \tag{77}$$

from which it can be seen that $S(t) = 0$ is a valid solution. Thus, the unique solution to Eq 77 consistent with the boundary condition $S(T) = 0$ is $S(t) = 0$ identically, indicating that our choice of cost functional eliminates any temporal transients in the Riccati differential equation, allowing us to replace it with the algebraic Riccati equation (Eq 53).

## Author Contributions

**Conceptualization:** Nadav Amir, Naftali Tishby, Israel Nelken.

**Data curation:** Reut Suliman-Lavie, Maayan Tal.

**Formal analysis:** Nadav Amir, Israel Nelken.

**Funding acquisition:** Sagiv Shifman, Israel Nelken.

**Investigation:** Nadav Amir, Reut Suliman-Lavie, Maayan Tal, Sagiv Shifman, Naftali Tishby, Israel Nelken.

**Methodology:** Nadav Amir, Naftali Tishby, Israel Nelken.

**Resources:** Sagiv Shifman, Naftali Tishby, Israel Nelken.

**Supervision:** Sagiv Shifman, Naftali Tishby, Israel Nelken.

**Visualization:** Nadav Amir.

**Writing – original draft:** Nadav Amir, Israel Nelken.

**Writing – review & editing:** Nadav Amir, Sagiv Shifman, Naftali Tishby, Israel Nelken.

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
