## [Decision Letter · Decision Letter 0]

19 May 2020

Dear Dr. Nelken,

Thank you very much for submitting your manuscript "Value-complexity tradeoff explains mouse navigational learning" for consideration at PLOS Computational Biology.

As with all papers reviewed by the journal, your manuscript was reviewed by members of the editorial board and by several independent reviewers. In light of the reviews (below this email), we would like to invite the resubmission of a significantly-revised version that takes into account all of the reviewers' comments.

Comparisons to other models and some additional analysis of the implications for theoretical neuroscience would be important to include. Also, please be sure to address the question of whether the code will be shared openly - this would be strongly encouraged.

We cannot make any decision about publication until we have seen the revised manuscript and your response to the reviewers' comments. Your revised manuscript is also likely to be sent to reviewers for further evaluation.

Sincerely,

Blake A. Richards

Associate Editor

PLOS Computational Biology

Samuel Gershman

Deputy Editor

PLOS Computational Biology

Comparisons to other models and some additional analysis of the implications for theoretical neuroscience would be important to include. Also, please be sure to address the question of whether the code will be shared openly - this would be strongly encouraged.

Reviewer's Responses to Questions

**Comments to the Authors:**

Reviewer #1: The manuscript describes a formal mathematical framework intended to capture the key features of learning to navigate a water maze by rodents, and applies it to model or fit behavioral data from wild-type and Pogz-gene-mutated mice. There are several components to the theoretical work reported here: (a) formulating a simplified model of the behavior of naïve animals, as a damped oscillator; (b) defining a cost function that combines the notion of favoring trajectories that lead to the submerged platform with the notion of adopting as simple a strategy as possible; (c) fitting the model to the data and extracting parameters that succinctly describe the learning process, in particular the inverse temperature Beta.

I admire the ambitious attempt to theorize about a behavior that many labs across the world have observed endless times, and for which there seems to be a need for a comprehensive theory. The theory developed here is undoubtedly elegant and mathematically well formulated, and it is abstract enough to generate that sense of mystical awe that is a necessary attribute of a psychologically satisfying theory. Personally, however, I am a bit skeptical about the implicit “behaviorist” assumption that either naïve rodents, with all their complexity and individual differences, or the learning process they demonstrate in the water maze, with its interaction with specific psychological factors, can be reduced to the basic physics of a harmonic oscillator or to the basic informatics of Kullback-Leibler minimization. If such a reduction were indeed possible, it would suggest that mice (at least when they learn to navigate a water maze) do not really need their brain, a standard algorithm is sufficient – which was after all the spirit in which behaviorism proceeded in the 50’s, before being replaced by cognitive neuroscience. Looking at data like that in Fig.5, with its wide scatter of individual data points over orders of magnitude, does not help quell my skepticism. Still, simplified models can be very useful even if they neglect important factors and only approximately fit the data, so I read the manuscript with this consideration in mind.

Component (a), the linear oscillator model of naïve trajectories in a water maze, is striking in its simplicity and fun to read about. I was underwhelmed, however, by the limited discussion of its suitability for the problem at hand. Yes, they say that a linear model is required by the formalism to be adopted, and that it is not intended to model individual trajectories, of course; but I would have expected to see more of a discussion of what the model misses out, what are the major collective and individual avenues of departure from the model, and to what extent one could imagine to account for them in an extended description, perhaps as a perturbation, to stay in the physics domain.

Component (b) is the core contribution of this study, and is summarized by the attractive notion that learning (in this very specific instance) proceeds by maximizing value for the learner while trying to minimize or at least to control the increase in its (behavioral?) complexity, where complexity is defined in a somewhat recursive fashion not based on intrinsic parameters of the behavior, but rather on how behavior departs, microscopically, from that of a naïve agent, who has not learnt a thing. The application to the data then leads to the stimulating suggestion that complexity might increase early in learning, and decrease in its final stages. To me, this is another under-discussed aspect of the study. First, is it a strong result? Second, what does it really hinge on? Would more standard measures of trajectory complexity also produce it? Third, how does it square with the discrepancy between model-based prediction and observation in the case of speed (Fig.3c)? Fourth, is it reflected at all in the monotonic trends in Fig.3a,b,d, or if not would those trends have resulted from basic value maximization, or perhaps assuming a more basic constraint that the complexity as defined here?

Component (c) is a key component of the study, which distinguishes it from too abstract an academic exercise, and I much appreciate it. I would have liked, however, to see more of a discussion of the specifics of the genetically modified mice used, and of what the analysis might reveal about them, also in relation to gender effects.

In conclusion, I am impressed by the mathematical sophistication and technical prowess of this study, and I would enjoy a deeper analysis of its implications for theoretical neuroscience.

Reviewer #2: Disclaimer: I am not an expert in the methods described in this paper but I have some familiarity with reinforcement learning models and quantifying the behaviour in rodent trajectories.

This paper describes a new normative model which quantifies rodent trajectories during the well-established spatial memory task of the Morris water maze. This approach uses dynamical systems theory from statistical physics. The authors compared predictions from the model with experimental data from wild type and transgenic mice lines (N=51).

The key contribution is observed trajectory data can be explained in a trade-off between two model-derived quantities. The value of a trajectory and the complexity of a trajectory. The value is the energetic cost and was found to monotonically increase throughout the four training days. The complexity is the likelihood of the trajectory of a naïve animal behaving stochastically. The complexity was first observed to increase as paths become more random, and then decrease as paths become simplistic. This model may be useful to researchers who are interested in understanding differences in performance across rodent types and experimental manipulations within and beyond the Morris’ water maze, and more broadly for trajectory analysis in many species. Thus this work provides a potentially useful advance.

Below are a list of suggestions for improvements for the manuscript:

1. A key weakness with the current manuscript is that absence of another model-fit to the data for comparison to the one proposed. Currently, the model shows interesting findings such as the increase in value of time and non-monotonic relationship to complexity. However, there is no comparison to alternative plausible models.

2. The authors state: “Mean velocity had the correct range of values, although it failed to show the predicted non-monotonic dependence on the trade-off parameter β (potentially because of the high variability of the experimental data).”. This point should be discussed more in the discussion. What might underlie the high variability in the data not captured in the model? Why would the variability specifically affect the predicted dependence? This isn’t a given and would be worth explaining more for readers.

3. There is another recent addition to reinforcement learning that also references the Todorov PNAS paper called linear reinforcement learning. It is also uses a quantity derived from the cost of a trajectory similar to the value described here and is analytically solvable. This work is currently on bioRxiv and not published as of yet (https://www.biorxiv.org/content/10.1101/856849v1.full.pdf). It would be important for the authors to comment on this and consider it in their discussion.

4. Please make clear if you are aiming to publish your code along with the publication. This is important especially for a computational paper where people may be able to make use of the advances made with this model. For instance using an online repository such as GitHub.

5. Figure 4 is worth re-working for greater clarity. The image shows a blurring of the arrow head with the colour scale for the arrow for the data, is this a range of arrows superimposed or information about the precision of the arrow? I would also be improved by providing second example at a different point in the learning process and marking S for the release point to match the legend in the images.

Reviewer #3: Review uploaded as attachment.

**Have all data underlying the figures and results presented in the manuscript been provided?**

Reviewer #1: No: I would like to see more trajectories, produced by mice and by the model

Reviewer #2: Yes

Reviewer #3: Yes

PLOS authors have the option to publish the peer review history of their article (what does this mean?). If published, this will include your full peer review and any attached files.

Reviewer #1: No

Reviewer #2: No

Reviewer #3: No
---

## [Decision Letter · Decision Letter 1]

14 Sep 2020

Dear Dr. Nelken,

Thank you very much for submitting your manuscript "Value-complexity tradeoff explains mouse navigational learning" for consideration at PLOS Computational Biology. As with all papers reviewed by the journal, your manuscript was reviewed by members of the editorial board and by several independent reviewers. The reviewers appreciated the attention to an important topic. Based on the reviews, we are likely to accept this manuscript for publication, providing that you modify the manuscript according to the review recommendations.

Two of the three reviewers are happy with the revisions you have made to your manuscript. However, the third reviewer feels strongly that the manuscript lacks explanatory and mathematical details. They have noted a number of specific areas in which the explication is lacking in clarity. Please be sure to address these specific issues, and also, be sure to give the paper a thorough editing with an eye towards the question of ease of reader understanding.

Sincerely,

Blake A. Richards

Associate Editor

PLOS Computational Biology

Samuel Gershman

Deputy Editor

PLOS Computational Biology

[LINK]

Reviewer's Responses to Questions

**Comments to the Authors:**

Reviewer #1: The authors have extensively and satisfyingly revised their manuscript in response to the comments by all three reviewers. Congratulations.

Reviewer #2: The authors have suitably addressed the questions raised in this review.

Reviewer #3: Review of the revised manuscript “Value-complexity tradeoff explains mouse navigational learning” by Amir et al. (PDF of the review report also uploaded):

While authors provide thorough and interesting responses to some of the points raised in the previous reviewer report (points 7-13), responses to other issues raised (points 1-6) are either lacunary or not reflected in the manuscript, or both. I thus find myself in the uncomfortable position of having to comment on a paper which I find quite beautiful and worthy, and whose publication I definitely recommend as it presents a new way of thinking about navigational behavior from an information theoretic and constrained optimization angle—but, at the same time, having to contend with responses by authors who do not seem to take the review process seriously nor wish to include necessary material in the manuscript.

For example:

1) In their response to point 2.a, the authors state that they “added some information at the introduction”—but nothing substantial has been included in the introduction: all I could find in connection with Sanov’s theorem is a mention that the complexity cost is “theoretically justified.” This will hardly be helpful to the many readers who don’t come with a robust mathematical background! The relation of the author’s complexity measure to large deviations theory should be presented comprehensively.

2) In response to points 2.b and c, the authors provide a helpful and interesting discussion in their reply, which would indeed be useful to readers of the paper—but it has not been included in the paper in any form, as far as I can tell.

3) In response to point 3.a, no explanation is provided by the authors about the way in which the dynamic programming (Bellman) problem is converted into local differential (or finite difference) equations. If the reader is to be expected to actually understand the approach proposed by the authors, this kind of explanation is necessary! Since the authors claim to be solving an optimization problem, indeed of the Bellman form, the reader should be able to tell why the Ricatti equations fulfill the job—and this even more so in a publication whose readership includes experimentalists. The authors should include a detailed discussion of the mathematics involved here, and of how one converts the Bellman problem into the Ricatti equations.

4) How boundary conditions and transient solutions are treated is an important aspect of the solution of the problem posed by the authors. The question of transient solutions, that arise because of temporal boundary conditions (point 3b), is of particular relevance. In this connection, the authors merely added a brief phrase in the revised manuscript (lines 839-840) that states that transient solutions are neglected. A detailed treatments of boundary conditions and transient solutions should be presented.

5) Figure 4 (point 5) is improved, but some lingering issues remain:

- the low values of beta are not apparent in the illustration (no blue arrow is visible);

- it not clear whether black arrows are closer to red or yellow arrows (both cases appear to occur in the diagram); it would make sense to show also arrows corresponding to the value of beta that best fits the data.

Finally, a minor issue is that, on line 654, Ref [34] is repeated.

In sum, while I find that this paper would be a wonderful addition to the literature, in my view publication would be warranted only if the authors greatly expand the mathematical and technical discussions pertaining to their way of defining and solving the problem, including the points mentioned above.

**Have all data underlying the figures and results presented in the manuscript been provided?**

Reviewer #1: Yes

Reviewer #2: Yes

Reviewer #3: Yes

PLOS authors have the option to publish the peer review history of their article (what does this mean?). If published, this will include your full peer review and any attached files.

Reviewer #1: **Yes: **Alessandro Treves

Reviewer #2: No

Reviewer #3: No
---

## [Editor Report · Decision Letter 2]

6 Nov 2020

Dear Dr. Nelken,

We are pleased to inform you that your manuscript 'Value-complexity tradeoff explains mouse navigational learning' has been provisionally accepted for publication in PLOS Computational Biology.

Best regards,

Blake A. Richards

Associate Editor

PLOS Computational Biology

Samuel Gershman

Deputy Editor

PLOS Computational Biology

---

## [Editor Report · Acceptance letter]

1 Dec 2020

PCOMPBIOL-D-20-00247R2 

Value-complexity tradeoff explains mouse navigational learning

Dear Dr Nelken,

I am pleased to inform you that your manuscript has been formally accepted for publication in PLOS Computational Biology. Your manuscript is now with our production department and you will be notified of the publication date in due course.

With kind regards,

Nicola Davies
